# BRCA2 controls DNA:RNA hybrid level at DSBs by mediating RNase H2 recruitment

Giuseppina D'Alessandro[1], Donna Rose Whelan [2], Sean Michael Howard[3], Valerio Vitelli[1], Xavier Renaudin[4], Marek Adamowicz[1,7], Fabio Iannelli[1], Corey Winston Jones-Weinert[1], MiYoung Lee[4], Valentina Matti[1], Wei Ting C. Lee[2], Michael John Morten[2], Ashok Raraakrishnan Venkitaraman[4], Petr Cejka[3,5], Eli Rothenberg[2] & Fabrizio d'Adda di Fagagna [1,6]

DNA double-strand breaks (DSBs) are toxic DNA lesions, which, if not properly repaired, may lead to genomic instability, cell death and senescence. Damage-induced long non-coding RNAs (dilncRNAs) are transcribed from broken DNA ends and contribute to DNA damage response (DDR) signaling. Here we show that dilncRNAs play a role in DSB repair by homologous recombination (HR) by contributing to the recruitment of the HR proteins BRCA1, BRCA2, and RAD51, without affecting DNA-end resection. In S/G2-phase cells, dilncRNAs pair to the resected DNA ends and form DNA:RNA hybrids, which are recognized by BRCA1. We also show that BRCA2 directly interacts with RNase H2, mediates its localization to DSBs in the S/G2 cell-cycle phase, and controls DNA:RNA hybrid levels at DSBs. These results demonstrate that regulated DNA:RNA hybrid levels at DSBs contribute to HR-mediated repair.

[1] IFOM, the FIRC Institute of Molecular Oncology, Via Adamello 16, Milan 20139, Italy. [2] Department of Biochemistry and Molecular Pharmacology, NYU School of Medicine, New York, NY 10016, USA. [3] Institute for Research in Biomedicine, Università della Svizzera italiana, Via Vela 6, Bellinzona 6500, Switzerland. [4] Medical Research Council Cancer Unit, University of Cambridge, Hills Road, Cambridge CB2 0XZ, UK. [5] Department of Biology, Institute of Biochemistry, Swiss Federal Institute of Technology, Otto-Stern-Weg 3, Zurich 8093, Switzerland. [6] Istituto di Genetica Molecolare, Consiglio Nazionale delle Ricerche (IGM-CNR), Via Abbiategrasso 207, Pavia 27100, Italy. [7] Present address: Genome Damage and Stability Centre, School of Life Sciences, University of Sussex, Falmer, Brighton BN1 9RH, UK. Correspondence and requests for materials should be addressed to G.D'A. (email: dalessandrogiuseppina@gmail.com) or to F.d'A.d F. (email: fabrizio.dadda@ifom.eu)

DNA double-strand breaks (DSBs) are some of the most toxic DNA lesions, since their inaccurate repair may result in mutations that contribute to cancer onset and progression, and to the development of neurological and immunological disorders[1]. The formation of DSBs activates a cellular response known as the DNA damage response (DDR), which senses the lesion, signals its presence, and coordinates its repair[2,3]. Following detection of DSB or resected DNA ends by the MRE11-RAD50-NBS1 (MRN) complex or the single-strand DNA binding protein replication protein A (RPA), respectively, apical kinases, such as ataxia-telangiectasia mutated (ATM) and ATM- and Rad3-related (ATR), are activated and phosphorylate numerous targets, including the histone variant H2AX (named γH2AX). The spreading of γH2AX along the chromosome favors the recruitment of additional DDR proteins, including p53-binding protein (53BP1) and breast cancer 1 (BRCA1), which accumulate in cytologically detectable DDR foci[4]. In mammalian cells, DSBs are mainly repaired by ligation of the broken DNA ends in a process known as nonhomologous end-joining (NHEJ)[5]. However, during the S/G2 cell-cycle phase, DSBs undergo resection, which directs repair toward homology-based mechanisms[6]. DNA-end resection is a process initiated by the coordinated action of the MRE11 nuclease within the MRN complex, together with C-terminal binding protein interacting protein (CtIP), and continued by the nucleases including exonuclease 1 (EXO1) or DNA2[7]. Resected DNA ends are coated by RPA, which contributes to DDR signaling and undergoes a DNA damage-dependent hyperphosphorylation[8]. When complementary sequences are exposed upon resection of both the DSB ends, RAD52 mediates their annealing via a process called single-strand annealing (SSA) resulting in the loss of genetic information[6]. Alternatively, a homologous sequence located on the sister chromatid or on the homologous chromosome can be used as a template for repair in a process known as homologous recombination (HR)[9]. The invasion of the homologous sequence is mediated by the recombinase RAD51, whose loading on the ssDNA ends is promoted by breast cancer 2 (BRCA2), which binds BRCA1 through the partner and localizer of BRCA2 (PALB2)[10,11]. BRCA1, together with its constitutive heterodimer BARD1, is a multifaceted protein with several roles in DDR signaling and repair[12]. BRCA1 and BRCA2 genes are the most frequently mutated genes in breast and ovarian cancers[13] and recently developed drugs, such as poly(ADP-ribose) polymerases (PARP) inhibitors, selectively target cancer cells harboring mutations in these genes[14]. Among its several functions, BRCA1 promotes DNA-end resection, mainly by counteracting the inhibitory effect of 53BP1[15]. Indeed, the HR defect in BRCA1-deficient cells is rescued by the depletion of 53BP1[16].

Recently, a novel role for RNA in the DNA damage signaling and repair has emerged[17–25]. In particular, we have reported that RNA polymerase II (RNA pol II) is recruited to DSBs, where it synthesizes damage-induced long noncoding RNAs (dilncRNAs)[17,18]. DilncRNAs are processed to generate DNA damage response RNAs (DDRNAs), which promote DDR signaling[17,18,21,25,26]. Similar RNA molecules, named diRNAs, contribute to DSB repair by HR[22–24].

It has recently been demonstrated that DNA:RNA hybrids form at DSBs in a tightly regulated fashion in *Schizosaccharomyces pombe*[27] and in mammalian cells[28–31]. However, DNA:RNA hybrid formation at DSBs in mammalian cells has not been investigated in depth yet, nor has any characterization of the molecular mechanisms leading to their formation or metabolism at DSBs been reported. Control of DNA:RNA hybrid levels can be achieved either by avoiding their formation during transcription, or by unwinding or degradation of already-formed hybrids by helicases and RNase H enzymes, respectively. In eukaryotic cells, DNA:RNA hybrids are degraded by RNase H1 and RNase H2, the latter accounting for the majority of RNase H activity in mammalian nuclei. RNase H2 is a heterotrimeric complex composed of a conserved catalytic subunit, called RNASEH2A, and auxiliary subunits RNASEH2B and RNASEH2C[32], which mediate the interaction with other proteins. RNase H2, in addition to its role in removing misincorporated ribonucleotides from genomic DNA[33,34], which have been recently described as a source of PARP-trapping DNA lesions[35], is also responsible for the resolution of DNA:RNA hybrids generated by RNA pol II during transcription[36]. Very recently, RNase H2 was also shown to regulate DNA:RNA hybrid levels at telomeres[37]. Furthermore, a physiologic role for human RNase H2 was uncovered through the discovery that mutations in any of its subunits cause the Aicardi–Goutieres syndrome, a neuroinflammatory disease associated with the chronic activation of the immune system in response to an excessive accumulation of aberrant forms of nucleic acids[38].

Strong links between HR proteins, RNA, and DNA:RNA hybrids have already been demonstrated and continue to emerge: BRCA1 interacts with several transcription and RNA-processing factors including RNA pol II[39,40], recognizes and promotes the processing of miRNA precursors[41], and mediates the recruitment of the DNA/RNA helicase SENATAXIN to gene terminators to avoid genome instability induced by DNA:RNA hybrid accumulation[42]. Depletion of BRCA1 or BRCA2 results in the accumulation of DNA:RNA hybrids globally[43,44] and specifically at promoter proximal sites of actively transcribed genes[45,46]. In addition, proteins involved in the fanconi anemia (FA) repair pathway are recruited to DNA damage sites via DNA:RNA hybrids to suppress hybrid-associated genomic instability[47,48]. Altogether these results suggest an emerging but yet undefined intimate relationship between DNA:RNA hybrids and the HR repair pathway.

Herein, we explore the links between RNA, DNA:RNA hybrid formation and metabolism, and HR. We show that dilncRNAs generated at DSBs contribute to the recruitment of the HR proteins BRCA1/BRCA2/RAD51 to DSBs. Specifically, in the S/G2 cell-cycle phase dilncRNAs pair to the resected DNA ends to form DNA:RNA hybrids, which are directly recognized by BRCA1. Moreover, we demonstrate that BRCA2 modulates DNA:RNA hybrid levels at DSBs by interacting with and mediating RNase H2 recruitment to DSBs. Combined, our data provide a mechanistic model for the emerging interplay between DNA:RNA hybrids and HR proteins.

## Results

**DNA:RNA hybrids form at DSBs.** Recently, we reported that RNA pol II is recruited to DSBs where it transcribes dilncRNAs bidirectionally starting from exposed DNA ends[17]. In the same experimental setup where dilncRNAs were characterized, we investigated whether they could form DNA:RNA hybrids. We induced a site-specific DSB by transfecting HeLa cells with the I-PpoI nuclease, whose nuclear localization is induced by 4-hydroxytamoxifen (4-OHT). Upon I-PpoI-mediated DSB generation within the weakly transcribed *DAB1* gene (Supplementary Fig. 1a), we monitored the formation of DNA:RNA hybrids by DNA:RNA hybrid immunoprecipitation (DRIP): briefly, non-crosslinked DNA:RNA hybrids were immunopurified with the specific S9.6 monoclonal antibody and analyzed by quantitative polymerase chain reaction (qPCR). We observed that DSB generation induces the formation of DNA:RNA hybrids peaking at ~1.5 kb and up to 3 kb from both sides of the DSB (Fig. 1a, b), consistently with the already reported dilncRNAs generated upon cutting[17]. Importantly, when cut samples were treated with RNase H, levels of DNA:RNA hybrids strongly decreased, demonstrating the specificity of the signal (Fig. 1b).

The accumulation of DNA:RNA hybrids at both sides of the DSB resembled the RNA Pol II-mediated de novo bidirectional transcription of dilncRNAs from the DSB[17], suggesting that dilncRNAs, rather than pre-existing RNA, such as a mRNA, are

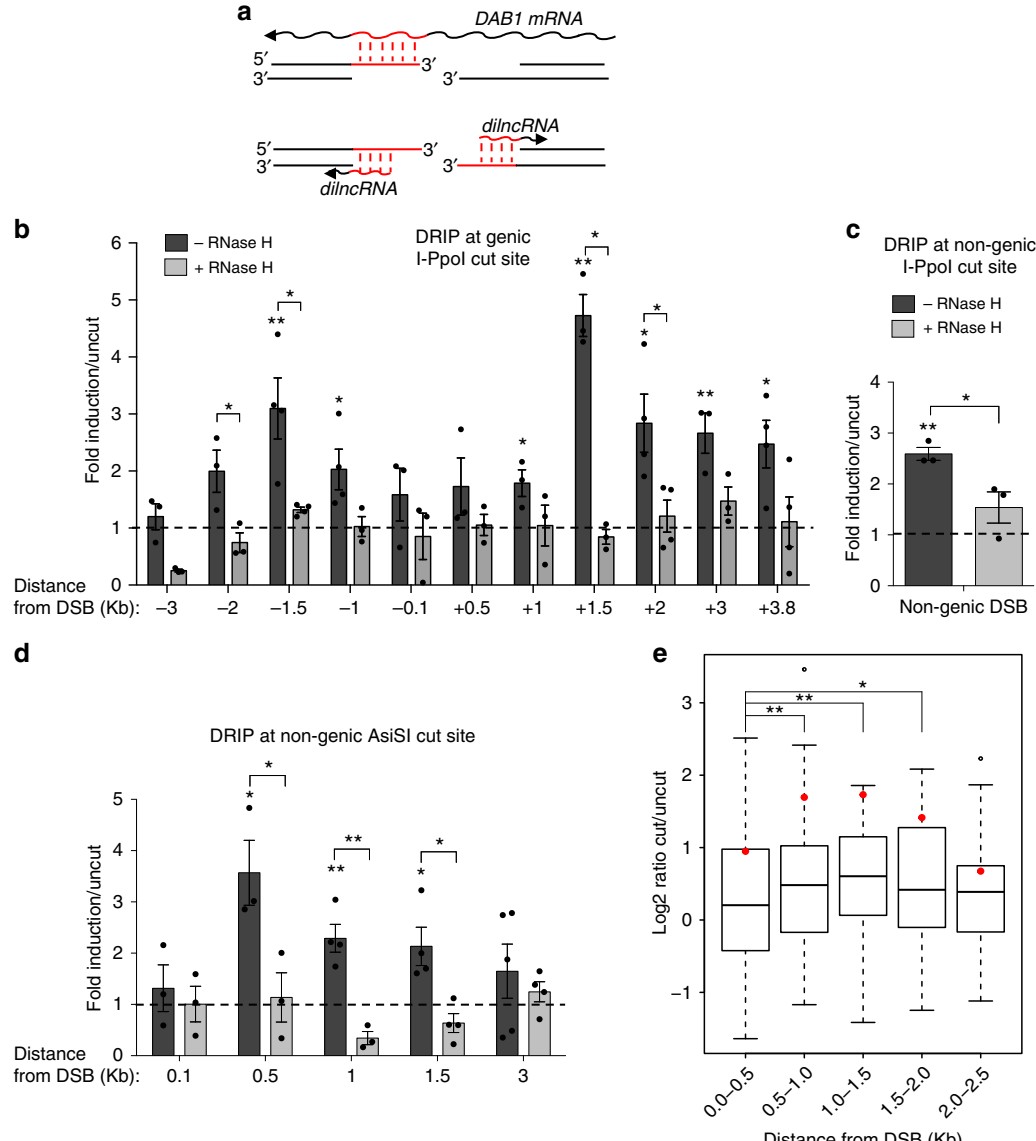

**Fig. 1** DNA:RNA hybrids form at DSBs independently of the genomic context. **a** Schematic representation of DNA:RNA hybrids (in red) that can be generated upon the hybridization of mRNA (top) or dilncRNAs (bottom) with resected DNA ends at the I-PpoI cut site within *DAB1* gene. **b** DRIP-qPCR analysis at the I-PpoI cut site within a genic (*DAB1* gene) or **c** nongenic locus in HeLa cells transfected with the I-PpoI nuclease. **d** DRIP-qPCR analysis at a nongenic AsiSI cut site in DIvA cells. Bar graphs in **b**, **c** and **d** show fold induction of DNA:RNA hybrid levels in cut samples relative to uncut. RNase H treatment was performed on cut samples to demonstrate specificity of the signal. Error bars represent s.e.m. ($n \geq 3$ independent experiments). $^{*}P < 0.05$, $^{**}P < 0.01$ (two-tailed Student's $t$ test). **e** Boxplot representing log2 ratio of the fold change of DNA:RNA hybrid reads in cut compared to uncut samples at the BLISS detected top 50 cut AsiSI sites at different distances from the DSBs. $^{*}P < 0.05$, $^{**}P < 0.01$ (Wilcoxon signed-rank test). Source data are provided as a Source Data file

generating the observed hybrids (Fig. 1a). To further confirm this observation, we monitored DNA:RNA hybrid accumulation at a DSB within a nongenic I-PpoI target site in HeLa cells (Supplementary Fig. 1a). Indeed, DSB induction by I-PpoI led to DNA:RNA hybrid accumulation also at this site (Fig. 1c), thus supporting the notion that DSBs induce the synthesis of de novo transcripts that form DNA:RNA hybrids at DSBs. In order to extend this observation to another nongenic region generated in a different cellular system in which DSBs are induced by a different nuclease, we used DIvA (DSB inducible via AsiSI) U2OS cells (Supplementary Fig. 1a), where nuclear localization of the AsiSI restriction enzyme is induced by 4-OHT to generate DSBs at distinct locations[49] (Supplementary Fig. 1b, c). By DRIP-qPCR analyses, we observed DNA:RNA hybrids accumulation at the

nongenic AsiSI-induced DSB analyzed (Fig. 1d). Consistently, strand-specific reverse transcription followed by qPCR confirmed dilncRNA accumulation upon damage at this nongenic AsiSI cleavage site (Supplementary Fig. 1d). Genome-wide accumulation and distribution of DNA:RNA hybrids at AsiSI-induced DSBs was recently investigated[30]. We further analyzed these DRIP-seq data and matched them with sites of AsiSI-induced DNA damage as determined in our laboratory by a high-resolution and genome-wide DSB mapping method named breaks labeling in situ and sequencing (BLISS)[50]. We analyzed the accumulation and enrichment of DNA:RNA hybrids in damaged compared to undamaged cells at efficiently cleaved AsiSI sites, as judged by BLISS, at different windows of distance from DSBs. This independent experiment and analysis

reconfirmed the DNA:RNA hybrid accumulation profile detected by DRIP-qPCR for the nongenic AsiSI site analyzed in Fig. 1d, shown as red dots in Fig. 1e. Most importantly, the analysis of the most efficiently cut AsiSI sites in this dataset indicates at the genome-wide level a general increase of DNA:RNA hybrid accumulation starting at 0.5 kb from the DSB (Fig. 1e). Additionally, DNA:RNA hybrid enrichment was observed both at DSBs located within genic and nongenic regions (Supplementary Fig. 1e), suggesting that DNA:RNA hybrids accumulate at sites of DNA damage independently from their transcribed or untranscribed status before DSB induction.

Since dilncRNAs production is dependent on RNA pol II[17,18], we tested whether RNA pol II inhibition with 5,6-dichloro-1-β-D-ribofuranosylbenzimidazole (DRB) affected DNA:RNA hybrid generation. DRIP-qPCR at the I-PpoI cleavage site within the DAB1 gene in HeLa cells treated with DRB, or with vehicle only, prior to DSB induction, revealed that DNA:RNA hybrid accumulation at the damaged site is dependent on RNA pol II (Supplementary Fig. 1f).

Collectively, these results show that DNA:RNA hybrids form at DSBs, as independently demonstrated site-specifically and genome-wide by DRIP analyses at genic and nongenic loci, and that their accumulation requires RNA pol II activity.

**DSB-induced DNA:RNA hybrids form at resected DNA ends.** Having demonstrated that DNA:RNA hybrids accumulate at I-PpoI- and AsiSI-induced DSBs regardless of the genomic location and that their formation requires RNA pol II activity, we reasoned that, during the S/G2 cell-cycle phase, DNA-end resection and consequent single-stranded DNA generation could provide a suitable DNA substrate for dilncRNA pairing to their resected template DNA and allow hybrids to form (Fig. 1a). We therefore tested whether DNA:RNA hybrid formation is modulated during the cell-cycle by using HeLa-FUCCI cells, which express the fluorescent ubiquitination-based cell-cycle indicators (FUCCI)[51]. Following I-PpoI expression, we sorted cells into G1- and S/G2-phase populations and we monitored DNA:RNA hybrid levels by DRIP-qPCR. Importantly, DNA:RNA hybrid accumulation was analyzed at 1.5 kb on the right from the I-PpoI-induced DSB within the DAB1 gene, where the resected DNA end could pair only with the newly synthesized dilncRNA and not with a potentially pre-existing mRNA (Fig. 1a). We observed that, upon DSB induction, DNA:RNA hybrids accumulate preferentially in the S/G2-phase of the cell-cycle (Fig. 2a).

We next aimed to extend our observations to DSBs formed throughout the genome by an independent approach. To that end, we utilized super-resolution fluorescence microscopy (STORM) and analyzed U2OS cells synchronized in G1- or S-phase and treated with the radiomimetic drug neocarzinostatin (NCS). We determined the extent of colocalization between the DDR marker γH2AX and DNA:RNA hybrids detected by the S9.6 antibody by quantifying the overlaps of their signals in each cell relative to the calculated number of overlaps present due to random distribution[52,53]. Importantly, the wider distribution of γH2AX compared to DNA:RNA hybrids was also accounted for in our analysis by this approach. We observed increased rates of colocalization between γH2AX and DNA:RNA hybrids in S- compared to G1-phase cells (Fig. 2b, c).

Given the preferential DNA:RNA hybrid accumulation at DSBs in the S/G2 cell-cycle phase, we tested the contribution of DNA-end resection to their formation. To this aim, we knocked-down EXO1 or CtIP and we monitored DNA:RNA hybrid levels by DRIP-qPCR at 1.5 kb on the right from the I-PpoI cleavage site within DAB1 gene, as above. We observed that inhibiting resection by knocking-down EXO1 (Supplementary Fig. 2a, b) impairs DNA:RNA hybrid accumulation (Fig. 2d), while not

affecting dilncRNA transcription (Fig. 2e). Knock-down of CtIP, which is required for MRN functions at DSBs, among which initiation of DNA-end resection (Supplementary Fig. 2c, d), reduces DNA:RNA hybrid (Fig. 2f) as well as dilncRNA levels (Fig. 2g), in line with the reported function of MRN in modulating dilncRNAs transcription[17]. In conclusion, while CtIP is required for dilncRNA transcription, probably by favouring other MRN functions rather than DNA-end resection, DNA-end resection per se, as shown by EXO1 knock-down, seems not to be required for dilncRNA transcription but it only provides available templates for dilncRNAs hybridization and, therefore, DNA:RNA hybrid formation. As a control, neither CtIP nor EXO1 knock-down altered cell-cycle phase distribution (Supplementary Fig. 2e).

Having observed that DNA:RNA hybrids accumulation occurs downstream of DNA-end resection, we aimed at monitoring the impact on transcriptional inhibition on DNA-end resection. In order to test this, we specifically and acutely inhibited RNA pol II activity with α-amanitin or DRB and we simultaneously irradiated (5 Gy) HeLa cells—effective RNA pol II inhibition was confirmed by monitoring by RT-qPCR c-FOS mRNA level, a specific RNA pol II transcript with a short half-life (Supplementary Fig. 3a, b). DNA-end resection was measured by immuno-fluorescence analyses of exposed ssDNA through native staining of incorporated BrdU, total RPA and its phosphorylated form (RPA2 pS4/8)—focal signals were quantified in S/G2 cells, as monitored by the S/G2 phase marker cyclin A. Consistently, all three markers revealed proficient DNA-end resection upon RNA pol II inhibition (Fig. 3a, b and Supplementary Fig. 3c), in line with the observed DNA:RNA hybrids accumulation downstream of DNA-end resection.

Overall, these data show that DNA:RNA hybrid formation occurs preferentially during the S/G2 cell-cycle phases and it is facilitated by DNA-end resection.

**dilncRNAs contribute to HR-mediated repair.** We next studied the impact of transcriptional inhibition on the focal accumulation of HR proteins, which are specifically recruited to DSBs in the S/G2 cell-cycle phase. We inhibited RNA pol II activity with α-amanitin or DRB and we simultaneously irradiated (5 Gy) HeLa cells, as described above. We observed that both treatments significantly impair the formation of BRCA1, BRCA2, and RAD51 foci (Fig. 3c, d and Supplementary Fig. 3d), despite unaltered overall protein levels (Supplementary Fig. 3e, f).

We next sought to test a direct role of dilncRNAs in HR downstream of DNA-end resection. For this, we employed the DR-GFP reporter cell system[54] in which HR between a mutated integrated GFP construct, containing the I-SceI recognition site, and a truncated GFP generates a functional GFP open reading frame (Supplementary Fig. 4a). Following I-SceI induction, HR can be monitored by either the evaluation of GFP expression by fluorescence-activated cell sorting (FACS) analysis in individual cells or, more directly but in bulk, by PCR amplification of the recombined genomic DNA sequence. We impaired dilncRNAs functions by complementary antisense oligonucleotides (ASOs) (Supplementary Fig. 4a)—ASOs are modified oligonucleotide widely used to inhibit the function of their target RNAs[55] and have been previously used by our group to target dilncRNAs and inhibit DDR[17,18]. We transfected different sets of ASOs complementary to the predicted dilncRNAs generated at the I-SceI locus and simultaneously induced I-SceI for 72 h. Both FACS analysis of GFP expression (Fig. 3e) and PCR to detect the recombined genomic locus (Fig. 3f) demonstrated that ASOs matching dilncRNAs reduce HR efficiency, while an ASO matching an unrelated sequence (CTRL) had no impact on HR (Fig. 3e, f). Importantly, the same ASOs inactivated by annealing with

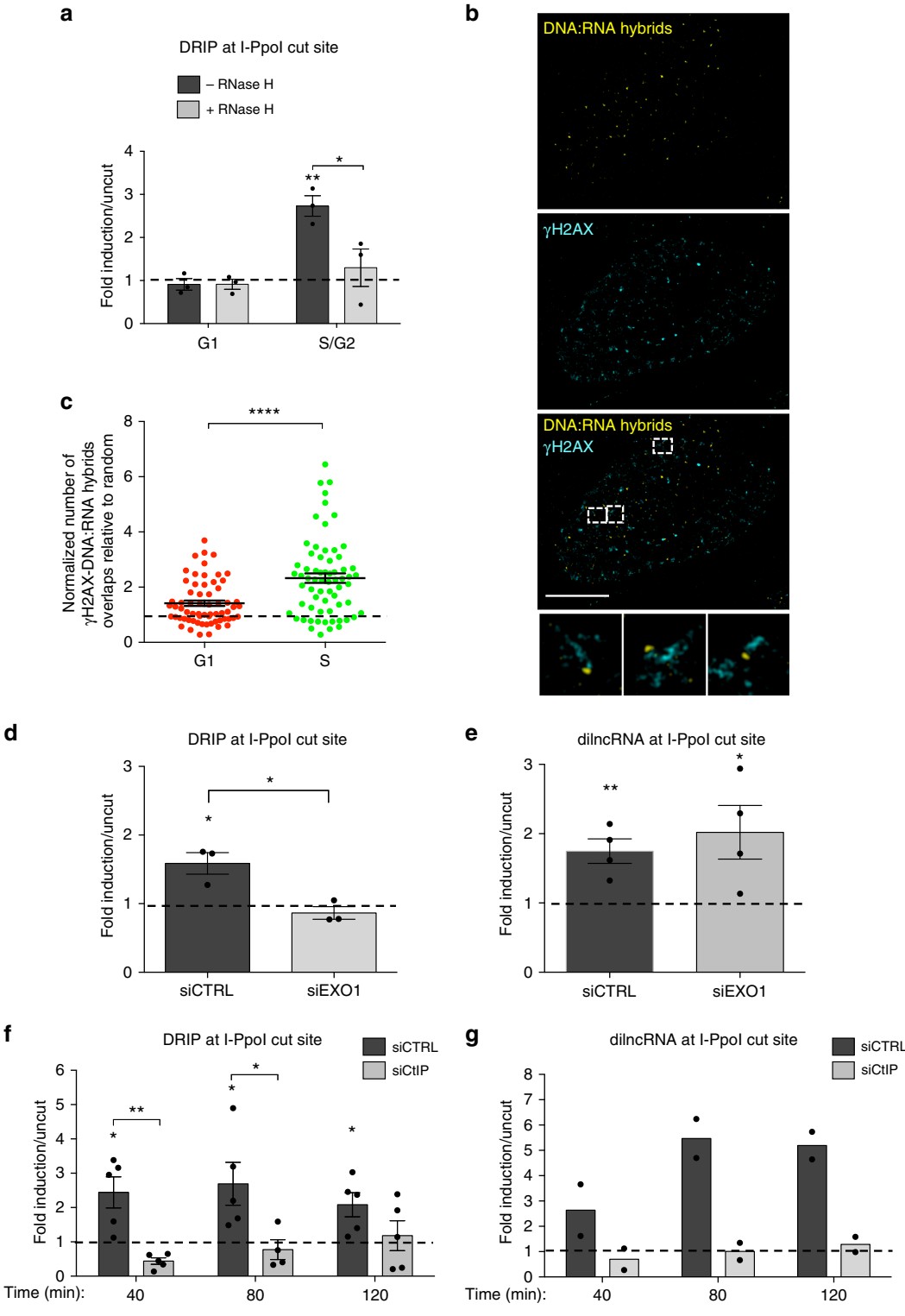

complementary sequences (Inactive) did not inhibit HR and all the ASOs left cell-cycle unaltered (Supplementary Fig. 4b). In this same cell system, genomic PCR can also be used to study SSA, a RAD52-dependent but RAD51-independent mechanism that shares with HR the initial DNA-end resection step. We observed that ASOs do not inhibit SSA (Fig. 3g), further indicating that dilncRNA inactivation impacts the HR process downstream of DNA-end resection. The impact of ASO-mediated dilncRNA inactivation on HR was further assessed using a different reporter system, the traffic

light reporter (TLR) system, that allows monitoring of both HR (GFP-positive cells) and mutagenic NHEJ (mCherry positive cells) events (Supplementary Fig. 4c)[56]. In this system, we confirmed that ASO-mediated dilncRNA inactivation reduces HR and we also observed an impairment of mutNHEJ, as monitored by FACS analysis of the percentage of GFP and mCherry positive cells, respectively (Supplementary Fig. 4d). Importantly, treatment with ASOs did not cause significant cell-cycle alterations (Supplementary Fig. 4e).

**Fig. 2** DSB-induced DNA:RNA hybrids form at resected DNA ends in S/G2-phase cells. **a** DRIP-qPCR analysis at 1.5 kb on the right from the I-PpoI cut site within *DAB1* gene in G1- or S/G2-phase-sorted HeLa-FUCCI cells transfected with the I-PpoI nuclease. The bar graph shows fold induction of DNA:RNA hybrid levels in cut samples relative to uncut. Error bars represent s.e.m. ($n = 3$ independent experiments). **b** Representative pictures of super-resolution imaging analysis of γH2AX (cyan) and DNA:RNA hybrids (yellow) colocalization in S-phase synchronized U2OS cells treated with neocarzinostatin (NCS). Scale bar: 5 μm. **c** Dot plot shows the normalized number of overlaps relative to random of γH2AX and DNA:RNA hybrids signals in G1- or S-phase NCS-treated U2OS cells. At least $n = 60$ events were counted from three independent experiments. Lines represent mean ± s.e.m. **d** DRIP-qPCR analysis at 1.5 kb on the right from the I-PpoI cut site within *DAB1* gene in cells knocked-down for EXO1. Error bars represent s.e.m. ($n = 3$ independent experiments). **e** Strand-specific RT–qPCR analysis of dilncRNAs levels at 1.5 kb on the right from the I-PpoI cut site within *DAB1* gene in cells knocked-down for EXO1. Error bars represent s.e.m. ($n = 4$ independent experiments). **f** DRIP-qPCR analysis at 1.5 kb on the right from the I-PpoI cut site within *DAB1* gene in cells knocked-down for CtIP at different time points after cut induction. Error bars represent s.e.m. ($n = 5$ independent experiments). **g** Strand-specific RT-qPCR analysis of dilncRNAs levels at 1.5 kb on the right from the I-PpoI cut site within *DAB1* gene in cells knocked-down for CtIP at different time points after cut induction ($n = 2$ independent experiments). $^*P < 0.05$, $^{**}P < 0.01$, $^{****}P < 0.0001$ (two-tailed Student's *t* test). Source data are provided as a Source Data file

These results show that BRCA1, BRCA2, and RAD51 focal accumulation to DSBs is reduced upon transcriptional inhibition and that ASOs-mediated dilncRNAs inactivation impairs HR.

**DNA:RNA hybrids are directly recognized by BRCA1.** Based on our observations that dilncRNAs form DNA:RNA hybrids in S/G2-phase cells and modulate the recruitment of HR proteins to sites of DNA damage, we sought to test the contribution of DNA:RNA hybrids to the focal accumulation of HR proteins at DSBs. By performing super-resolution imaging and analyzing the extent of colocalization between BRCA1 and DNA:RNA hybrids in NCS-treated U2OS cells synchronized in S-phase, we observed that the few detectable DNA:RNA hybrids often co-localize with BRCA1 in S-phase cells upon damage (Fig. 4a, b).

To test whether BRCA1 can directly recognize DNA:RNA hybrids, we used purified recombinant human BRCA1 or the constitutive BRCA1-BARD1 heterodimer in an electrophoretic mobility shift assay (EMSA) with either DNA duplexes or DNA:RNA hybrids. Radioactively labeled probes were incubated with the recombinant proteins and separated by electrophoresis on a native polyacrylamide gel. Both BRCA1 alone and BRCA1-BARD1 bound the DNA:RNA hybrid, with an affinity comparable to that for dsDNA (Fig. 4c, d). BRCA1-BARD1 binding with DNA:RNA hybrids, as well as dsDNA, was resistant to increasing salts concentrations, suggesting a robust binding (Supplementary Fig. 5a). Additionally, we performed competitive EMSA in which we prebound BRCA1-BARD1 with either labeled dsDNA or DNA:RNA hybrids and we then challenged the interaction with unlabeled DNA:RNA hybrids or dsDNA, respectively. In this experimental setup, we observed a modest preference for binding to hybrids, as opposed to dsDNA (Supplementary Fig. 5b).

Having observed that BRCA1 can bind DNA:RNA hybrids, we tested whether the modulation of DNA:RNA hybrid level at DSBs in living cells impacted on BRCA1 focal accumulation and/or retention at DSBs. To this purpose, we monitored BRCA1 foci formation at DSBs in irradiated (2 Gy) U2OS cells expressing RNase H1 fused to GFP, or GFP alone as a control. We observed that RNase H1 overexpression impairs ionizing radiation-induced BRCA1 foci formation (Fig. 4e, f), indicating a role for DNA:RNA hybrids in favouring BRCA1 recruitment and/or retention to DSBs. To rule out any indirect effect of RNase H1 overexpression, we treated irradiated cells with RNase H in situ. Briefly, we irradiated U2OS cells (2 Gy) and 1 h later we gently permeabilized and incubated them with recombinant bacterial RNase H. After 30 min, cells were fixed and BRCA1 foci were monitored in S/G2-phase cells. We observed that RNase H treatment reduces the amount of BRCA1 foci (Fig. 4g, h), while not impacting neither on the number of γ-H2AX foci (Supplementary Fig. 5c), nor on DNA-end resection, as determined by RPA foci (Supplementary Fig. 5d).

These results show that DNA:RNA hybrids can be directly recognized by BRCA1 in vitro and in living cells.

**RNase H2 is recruited to DSBs in the S/G2 cell-cycle phase.** The observation that excessive DNA:RNA hybrid accumulation may be detrimental for HR[29] could suggest that their levels at DSBs need to be tightly controlled. Since RNase H2 is the major source of RNase H activity in mammalian nuclei[57], we tested its recruitment to DSBs by performing chromatin immunoprecipitation (ChIP) and assaying for RNASEH2A enrichment at the nongenic AsiSI cut site in DIvA cells in which we described DNA:RNA hybrid accumulation upon DSB formation. We observed enrichment of RNASEH2A, and of γH2AX as a control, at the AsiSI cleavage site in cut relative to uncut cells (Fig. 5a and Supplementary Fig. 6a). To further validate this result with a different technique and at multiple genomic sites, we performed immunofluorescence microscopy to detect RNase H2 in irradiated or not irradiated cells. However, such stainings showed a diffuse signal that failed to reveal discrete foci under the conditions tested (Supplementary Fig. 6b). To increase the sensitivity and specificity of the signal, we performed proximity ligation assay (PLA) between RNASEH2A and γH2AX in not irradiated or irradiated (2 Gy) U2OS cells fixed 1 or 6 h after irradiation. We observed an increase in PLA signals between RNASEH2A and γH2AX in irradiated cells (Fig. 5b, c), thus suggesting that the two proteins become in close proximity upon irradiation. As a negative control, no signal was detected when only one of the two primary antibodies was used (Supplementary Fig. 6c) and, as a reference, a comparable PLA signal was observed between γH2AX and the HR marker RAD51 (Supplementary Fig. 6d). Since damage-induced DNA:RNA hybrids preferentially accumulate in the S/G2 cell-cycle phase, we monitored RNase H2 recruitment to DSBs in irradiated (2 Gy) and not irradiated HeLa-FUCCI cells. In this setup, we observed an increase in PLA signals between γH2AX and RNASEH2A in S/G2-phase irradiated cells compared to irradiated G1 or to S/G2 not irradiated cells (Fig. 5d, e), indicating that RNase H2 localizes to DSBs preferentially in the S/G2 cell-cycle phase. Similar results were obtained with a different antibody raised against RNASEH2A (Supplementary Fig. 6e) or the RNASEH2B subunit (Supplementary Fig. 6f). These conclusions were not biased by cell-cycle variations of γH2AX foci or of RNase H2 protein levels, as both the number of γH2AX foci (Supplementary Fig. 6g) and the RNASEH2A and RNASEH2B pan-nuclear signals (Supplementary Fig. 6h, i) remained unchanged in G1- versus S/G2-phase cells. The observed increased PLA signal in S/G2- compared to G1-phase cells, with unaltered levels of both antigens, further demonstrates the specificity of the assay and of the conclusions reached. In line with the S/G2-phase recruitment of RNase H2 to DSBs, we observed increased PLA signals between RNASEH2A and RPA upon ionizing radiation (Supplementary Fig. 6j), while no changes upon DNA damage induction were observed when PLA was performed between BRCA2 and RNASEH2A (Supplementary Fig. 6k), suggesting a potentially constitutive interaction between

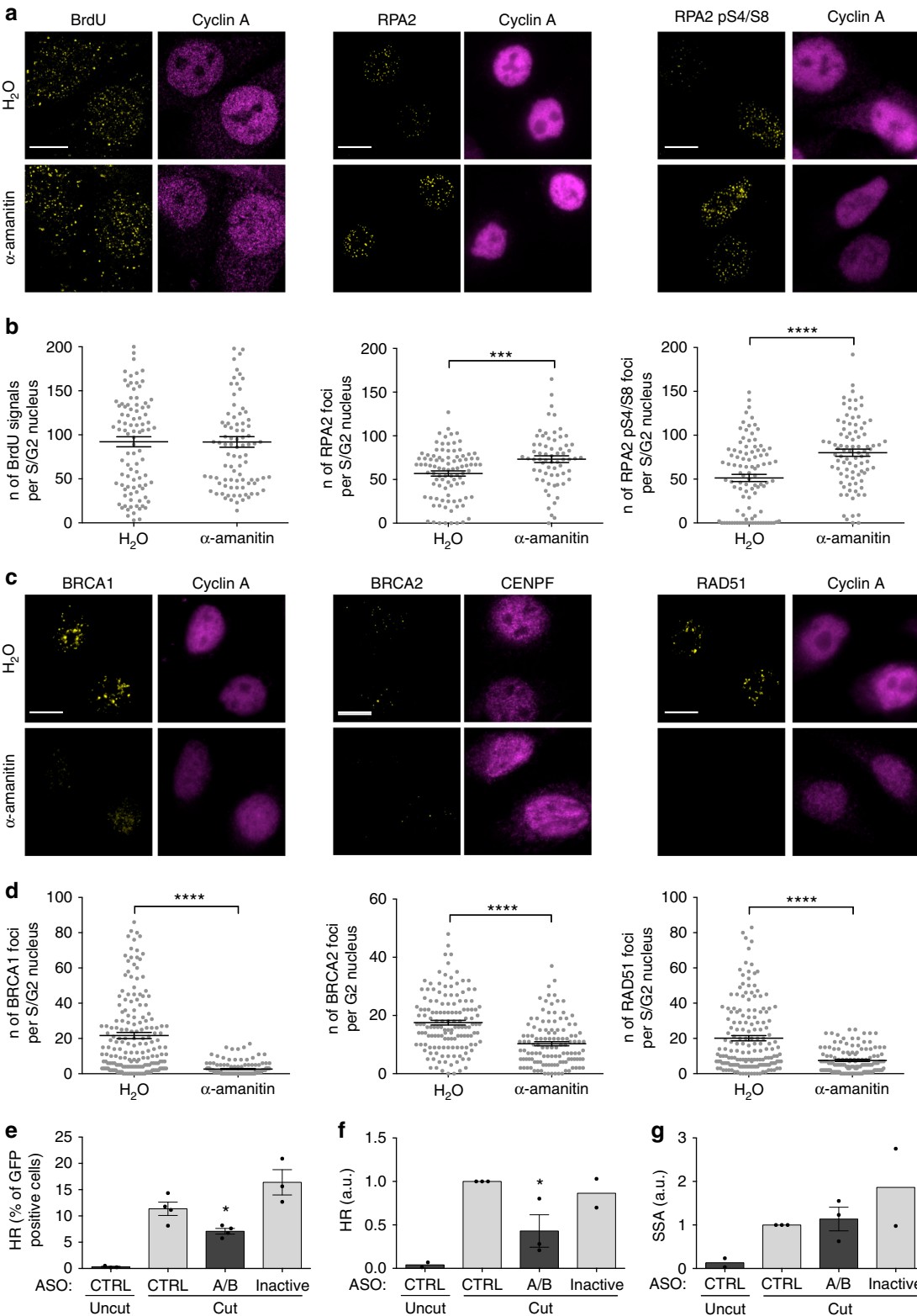

these two proteins. As a negative control, PLA between RNA-SEH2A and the centromeric protein B generated a very low signal, which did not increase upon DNA damage induction (Supplementary Fig. 6l).

In order to further extend our observations with an independent approach, we performed super-resolution imaging analysis of γH2AX and RNASEH2A colocalization in U2OS cells treated with NCS and we measured the extent of colocalization relative to random events. In agreement with PLA results, we observed that RNase H2 co-localizes with γH2AX in NCS-treated S-phase cells (Fig. 5f, g).

Overall, these results consistently indicate that RNase H2 is recruited to DSBs, both induced at a specific locus and genome-wide, preferentially during the S/G2 phase of the cell-cycle.

**Fig. 3** dilncRNAs contribute to HR proteins recruitment to DSBs and HR-mediated repair. **a** Representative images of DNA-end resection markers: ssDNA (visualized by BrdU native staining), RPA2, and RPA2 pS4/8 foci, co-stained with cyclin A, as S/G2-phase marker, in irradiated (5 Gy) HeLa cells treated with $H_2O$ or α-amanitin. Scale bar: 10 μm. **b** Dot plots show the number of signals/foci in **a**. At least $n = 60$ cells were counted from three independent experiments. Lines represent mean ± s.e.m. **c** Representative images of BRCA1, BRCA2, and RAD51 foci co-stained with cyclin A or CENPF, as S/G2-phase markers, in irradiated (5 Gy) HeLa cells treated with $H_2O$ or α-amanitin. Scale bar: 10 μm. **d** Dot plots show the number of foci in **c**. At least $n = 120$ cells were counted from three independent experiments. Lines represent mean ± s.e.m. **e–g** DR-GFP cells are treated with control ASO (CTRL), ASOs matching dilncRNAs (A/B), or inactive ASOs. **e** HR efficiency is monitored by FACS analysis of the percentage of GFP-positive cells or **f** by genomic semiquantitative PCR with primers P1 and P2 that only amplify the recombined GFP sequence generated after HR (see Supplementary Fig. 4a). **g** SSA efficiency is assessed by monitoring the 0.8 kb amplicon generated by genomic semiquantitative PCR with primers F1 and R2 (see Supplementary Fig. 4a). Bar graphs show the mean of $n \geq 2$ independent experiments. Error bars represent s.e.m. $^*P < 0.05$, $^{**}P < 0.01$, $^{***}P < 0.001$, $^{****}P < 0.0001$ (two-tailed Student's $t$ test). Source data are provided as a Source Data file

**BRCA2 and RNase H2 control DNA:RNA hybrid levels at DSBs.** Published reports suggest BRCA2 as a possible regulator of the cellular levels of DNA:RNA hybrids[43,46]. To test whether DNA:RNA hybrid levels at DSBs could be controlled by BRCA2 through RNase H2 recruitment, we monitored by super-resolution microscopy the colocalization of γH2AX and RNA-SEH2A in S-phase synchronized cells knocked-down for BRCA2 and treated with NCS. We observed that BRCA2 knock-down strongly reduces the colocalization of γH2AX and RNASEH2A (Fig. 6a). The same results were confirmed by PLA between γH2AX and RNASEH2A in S/G2-phase irradiated (2 Gy) HeLa-FUCCI cells knocked-down for BRCA2, which revealed reduced RNASEH2A recruitment to DSBs in the absence of BRCA2 (Fig. 6b and Supplementary Fig. 7a–c) despite no significant differences in the number of γH2AX foci (Supplementary Fig. 7d). The observed requirement of BRCA2 for RNase H2 recruitment to DSBs prompted us to test whether the two proteins could form a complex. We thus performed immunoprecipitation experiments from cell lysates of irradiated and not irradiated HEK293T cells prepared in the presence of benzonase to degrade all contaminating nucleic acids. We observed that RNASEH2A co-immunoprecipitates with BRCA2 and other proteins of the HR machinery, including BRCA1, PALB2, and RAD51, independently of DNA damage induction (Fig. 6c). This interaction is specific within this complex since no interactions with proteins known to be part of other BRCA1 complexes, such as CtIP and RAP80, were observed (Supplementary Fig. 7e). Having observed that BRCA2 is in a complex with and controls RNase H2 recruitment to DSBs, we tested whether the two proteins directly interact by performing pull-down experiments with a set of 9 purified recombinant GST-tagged BRCA2 fragments spanning its entire length[58] with recombinant purified RNase H2 complex. We observed that BRCA2 and RNase H2 directly interact through the BRC-containing region of BRCA2 (Fig. 6d). To additionally identify with more precision whether and which of the individual(s) BRC-repeats directly interact with RNase H2, we performed a pull-down of the 8 biotinylated BRC-peptides with the recombinant RNase H2 and we determined a specific interaction with BRC repeat 1 and 3 (Fig. 6e).

In order to test whether the impaired RNase H2 localization to DSBs upon BRCA2 inactivation resulted in increased DNA:RNA hybrid levels at DSBs, we monitored by super-resolution microscopy the colocalization of γH2AX and DNA:RNA hybrids in S-phase synchronized cells knocked-down for BRCA2 and treated with NCS. We observed that BRCA2 knock-down further increases the DNA:RNA hybrid signals at γH2AX foci (Fig. 6f). The same results were confirmed by DRIP-qPCR at the I-PpoI site within the *DAB1* gene in S/G2-phase-sorted HeLa-FUCCI cells knocked-down for BRCA2—sorting of the S/G2-phase cells population was necessary since BRCA2 inactivation affects the cell-cycle. DRIP-qPCR analysis revealed a significantly increased accumulation of DNA:RNA hybrids at the DSB in the absence of BRCA2 (Fig. 6g and Supplementary Fig. 7a), indicating that BRCA2, likely via the recruitment of RNase H2, regulates DNA:

RNA hybrid levels at DSBs. Interestingly, RAD51 knock-down in the same experimental conditions did not significantly alter DNA:RNA hybrid levels at the tested DSB (Supplementary Fig. 7a, f).

Altogether, these results show that BRCA2 directly interacts with RNase H2 and controls DNA:RNA hybrid levels at DSBs by mediating RNase H2 recruitment.

Finally, to determine the sequence of events at DSBs, we performed a series of ChIP to generate a time-course analysis of the recruitment of RNase H2 and key HR factors to DSB. To this end, we induced AsiSI cleavage in DIvA cells and we collected cells 2, 4, 6, and 8 h after DSB induction. We observed that γH2AX and BRCA1 signals are detectable already at early time points after DSB formation and are maintained throughout time points analyzed (Fig. 7a, b and Supplementary Fig. 7g, h). DNA-end resection, as monitored by RPA enrichment (Fig. 7c and Supplementary Fig. 7i), and BRCA2 recruitment (Fig. 7d and Supplementary Fig. 7j) were detectable 4 h upon DSB formation, followed by RNase H2 (Fig. 7e and Supplementary Fig. 7k) and RAD51 (Fig. 7f and Supplementary Fig. 7l), peaking at 6 and 8 h, respectively.

Together, these data support a model in which RNase H2 recruitment to DSBs is mediated by BRCA2 and is followed by RAD51 filament formation (Fig. 7g).

## Discussion

We have recently demonstrated that in mammalian cells RNA pol II is recruited to exposed DNA ends upon breakage, where it bidirectionally transcribes RNA species named dilncRNAs[17]. In the present study, we show that dilncRNAs form DNA:RNA hybrids downstream of DNA-end resection, upon hybridization with resected DNA ends, and contribute to HR. DSB-induced DNA:RNA hybrids are recognized by BRCA1 and their levels are modulated by BRCA2-mediated RNase H2 recruitment to DSBs.

Our observation that DNA:RNA hybrids form at DSBs in mammalian cells is in line with recent data in *S. pombe* and mammalian cells showing DNA:RNA hybrid accumulation at DSBs[27,28,30,31] and with the observed localization of the human RNA-unwinding protein DEAD box 1 (DDX1)[29] and the DNA/RNA helicase SENATAXIN[30,59] to DSBs in a transcription- and DNA:RNA hybrids-dependent manner. Importantly, we demonstrate that DNA:RNA hybrids form at DSBs in both genic and nongenic regions, in line with the observed DNA:RNA hybrid accumulation in regions both transcriptionally active and inactive prior to DNA damage induction[30], thus suggesting that newly transcribed dilncRNA could contribute to DNA:RNA hybrids formation. Further supporting this observation, we also demonstrate that the presence of resected DNA ends is required for DNA:RNA hybrids accumulation at DSBs. This indicates that DNA:RNA hybrid formation, even in genic regions, cannot only be the result of pairing of the pre-existing mRNA to the template DNA, since it would occur only on one side of the DSB: the one with exposed ssDNA matching the pre-existing transcript (see Fig. 1a). Conversely, the observed DNA:RNA hybrid accumulation at both sides of DSBs is compatible with newly

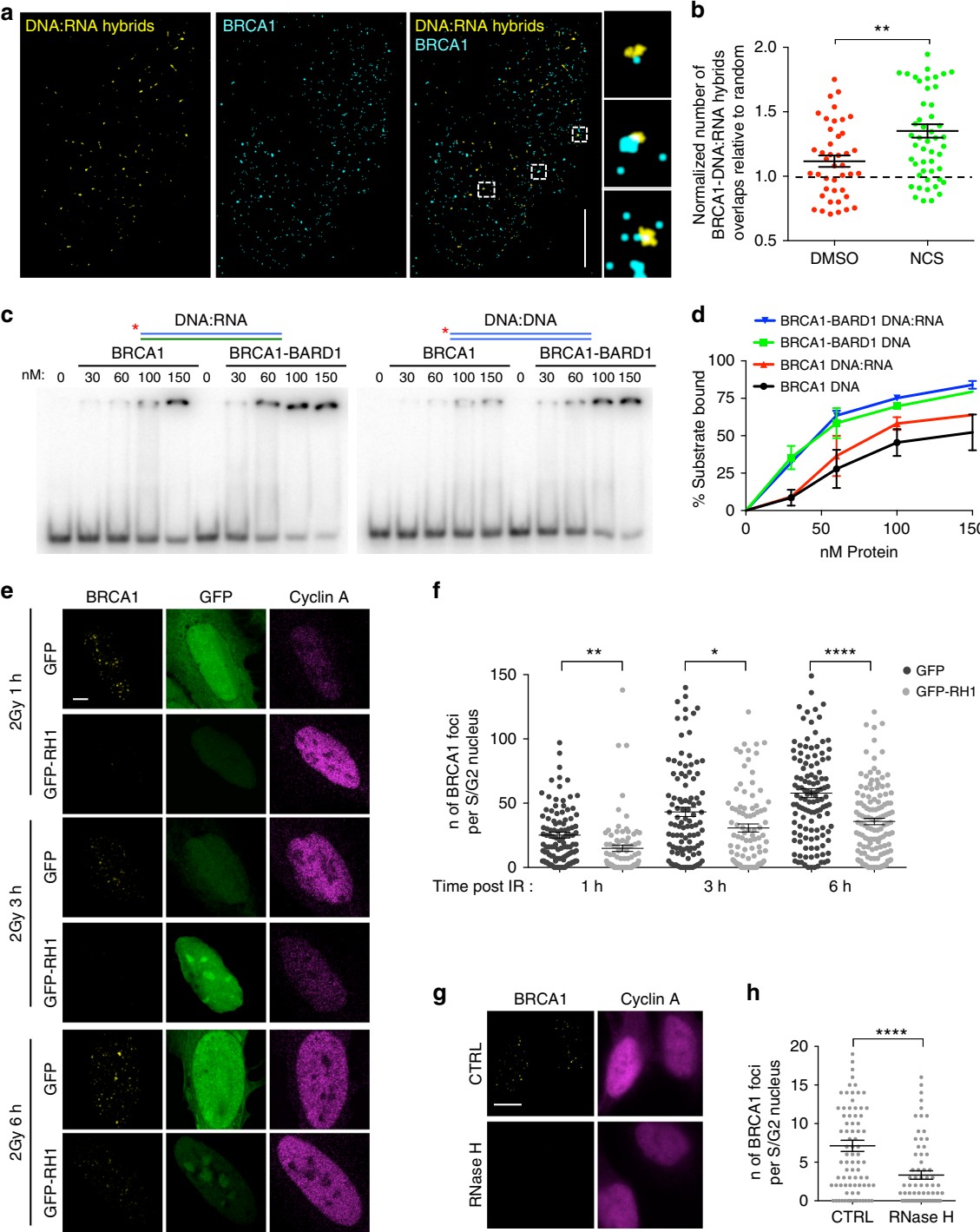

**Fig. 4** DNA:RNA hybrids are directly recognized by BRCA1 in vitro and in vivo. **a** Representative pictures of super-resolution imaging analysis of BRCA1 (cyan) and DNA:RNA hybrids (yellow) colocalization in S-phase synchronized NCS-treated U2OS cells. Scale bar: 5 μm. **b** Dot plot shows the normalized number of overlaps relative to random of BRCA1 and DNA:RNA hybrids signals in S-phase U2OS cells treated with DSMO or NCS. At least *n* = 40 events were counted from three independent experiments. Lines represent mean ± s.e.m. **c** Electrophoretic mobility shift assay (EMSA) of purified recombinant human BRCA1 or BRCA1-BARD1 with end-labeled (*) double-stranded DNA or DNA:RNA substrates. **d** Graph showing the percentage of protein-bound substrate at respective protein concentrations. Error bars represent s.e.m. (*n* = 2 independent experiments). **e** Representative images of BRCA1 foci co-stained with cyclin A, as S/G2-phase marker, in irradiated (2 Gy) U2OS cells over-expressing GFP or GFP-RNase H1 (GFP-RH1). Scale bar: 5 μm. **f** Dot plot shows the number of foci in **e**. At least *n* = 80 cells were counted from at least three independent experiments. Lines represent mean ± s.e.m. **g** Representative images of BRCA1 foci co-stained with cyclin A, as S/G2-phase marker, in irradiated (2 Gy) U2OS cells treated with RNase H prior to fixation. Scale bar: 10 μm. **h** Dot plot shows the number of foci in **g**. At least *n* = 80 cells were counted from three independent experiments. Lines represent mean ± s.e.m. *P < 0.05, **P < 0.01, ****P < 0.0001 (two-tailed Student's *t* test). Source data are provided as a Source Data file

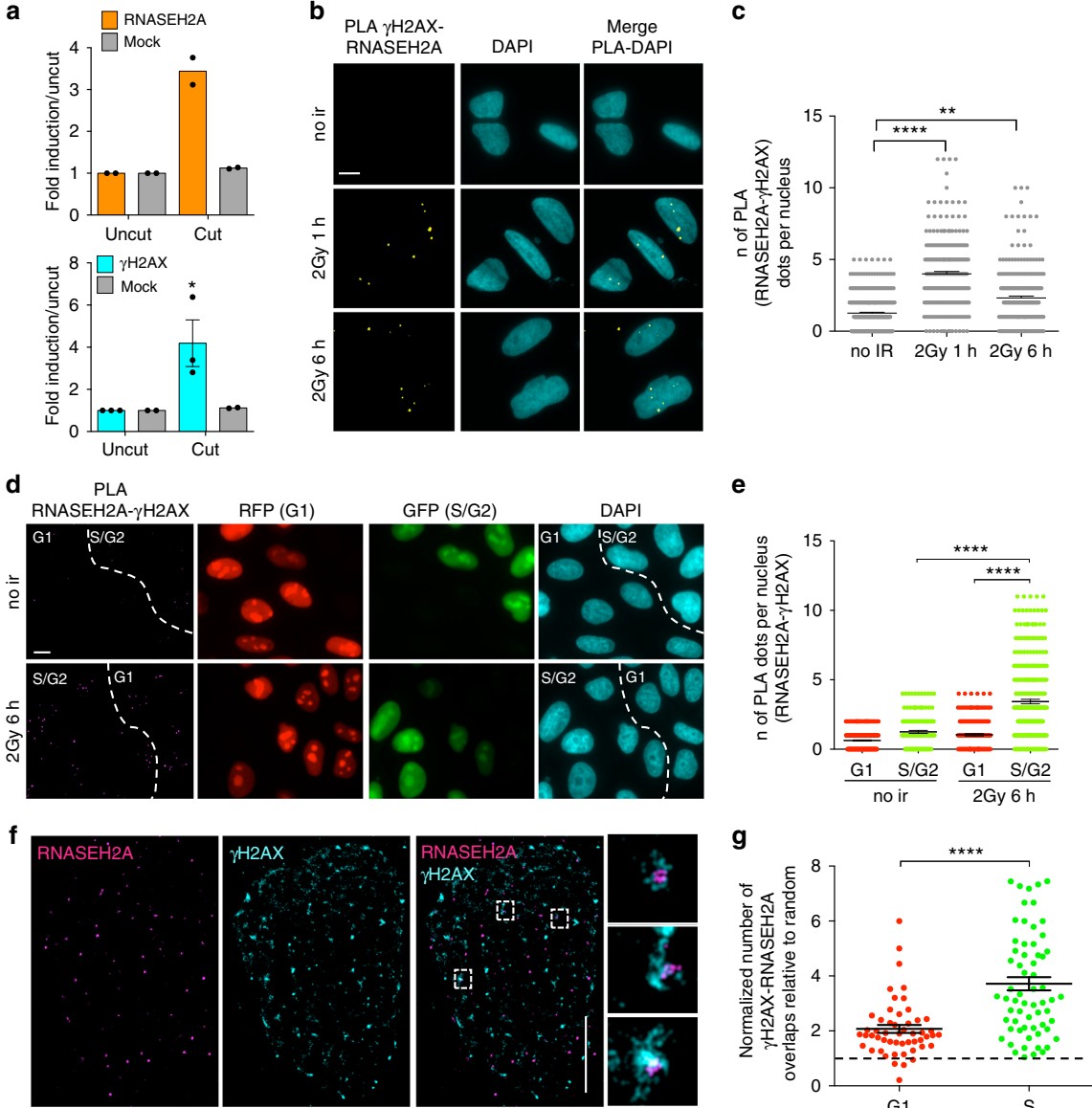

**Fig. 5** RNase H2 is recruited to DSBs preferentially in the S/G2-phase of the cell cycle. **a** ChIP of RNASEH2A (top) and γH2AX (bottom) at the AsiSI cut site in uncut or cut DIvA cells. The bar graph shows the fold induction in cut cells (6 h after AsiSI induction) compared to uncut. Error bars represent s.e.m. (n ≥ 2 biological replicates). **b** Representative images of PLA between RNASEH2A and γH2AX in not irradiated (no ir) or irradiated (2 Gy) U2OS cells. Scale bar: 10 μm. **c** Dot plot shows the number of signals per nucleus of PLA between RNASEH2A and γH2AX in not irradiated (no ir) or irradiated (2 Gy) U2OS cells. At least n = 200 cells were counted from four independent experiments. Lines represent mean ± s.e.m. **d** Representative images of PLA between RNASEH2A and γH2AX in not irradiated (no ir) or irradiated (2 Gy) HeLa-FUCCI cells. Scale bar: 10 μm. **e** Dot plot shows number of signals per nucleus of PLA between RNASEH2A and γH2AX in not irradiated (no ir) or irradiated (2 Gy) HeLa-FUCCI cells. At least n = 170 cells were counted from three independent experiments. Lines represent mean ± s.e.m. **f** Representative pictures of super-resolution imaging analysis of γH2AX (cyan) and RNASEH2A (magenta) colocalization in G1- or S-phase synchronized NCS-treated U2OS cells. Scale bar: 5 μm. **g** Dot plot shows the normalized number of overlaps relative to random of γH2AX and RNASEH2A signals in G1- or S-phase cells. At least n = 50 events were counted from three independent experiments. Lines represent mean ± s.e.m. **P < 0.01, ****P < 0.0001 (two-tailed Student's t test). Source data are provided as a Source Data file

bidirectionally transcribed dilncRNAs pairing with their template resected DNA ends. The DNA:RNA hybrids formation downstream of DNA-end resection is consistent with data generated in *S. pombe*[27] and with the observed requirement of DNA-end resection for the DNA:RNA hybrid-dependent recruitment of DDX1[29]. Interestingly, the reported need for pre-existing transcription to promote RNA-mediated repair in *Saccharomyces cerevisiae*[60] is consistent with the reported lack of recruitment of RNA polymerase II to DSBs and the lack of transcriptional induction in this species[61].

Our work also shows that transcriptional inhibition, while not reducing DNA-end resection, impairs the focal accumulation of the HR proteins BRCA1, BRCA2, and RAD51 at DSBs. Importantly, the specificity of the RNA Pol II inhibitors used (α-amanitin and DRB) makes unlikely the contribution of other RNA synthetizing enzymes to this process. In line with the reduced focal accumulation of HR proteins at DSBs upon RNA pol II inhibition, site-specific inactivation of dilncRNAs by complementary ASOs inhibits repair by HR, but does not affect SSA, which requires extensive DNA-end resection but differs from HR

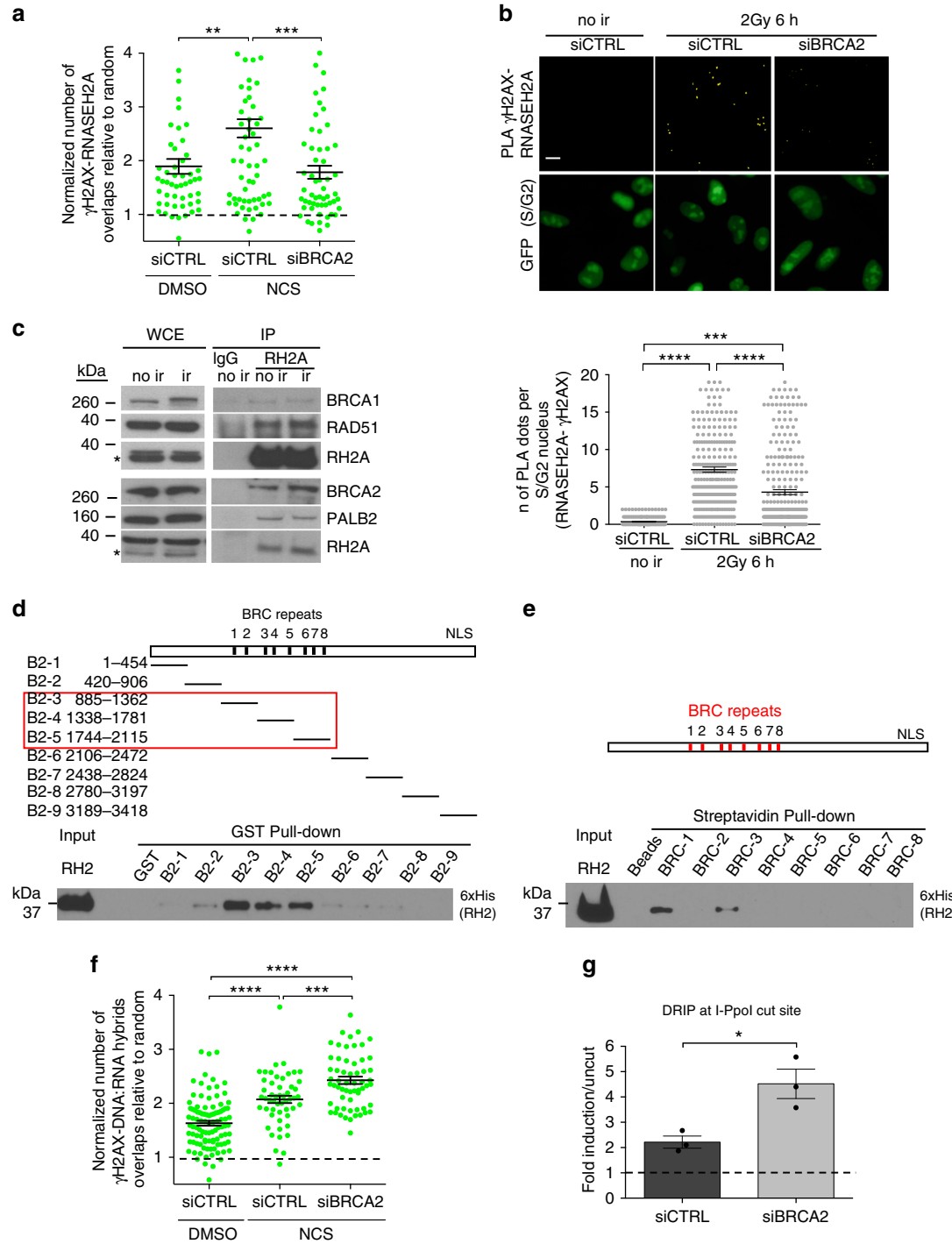

in the subsequent steps. The observation that transcriptional inhibition does not reduce DNA-end resection while impairing BRCA1 foci formation can be explained by the concomitant reduction of 53BP1 foci formation upon transcriptional inhibition or ASO treatment[17], which is also consistent with the observed reduction in NHEJ efficiency upon ASOs-mediated dilncRNAs inactivation. Indeed, since BRCA1 is required to oppose the inhibitory effect of 53BP1 on DNA-end resection, in the absence of 53BP1, as upon RNA pol II inhibition or ASO treatment, BRCA1 may become dispensable for this process[15]. This could also explain the observed stronger impact of transcriptional inhibition on BRCA1 recruitment to DSBs compared to BRCA2 and RAD51. Notably, the moderate increase of DNA-

end resection observed upon transcriptional inhibition may be caused either by a higher efficiency of the resection process, or, more intriguingly and consistent with our model, by an increased availability of single-stranded DNA for RPA binding in the absence of a competing complementary RNA paired to the resected DNA end.

At DSBs BRCA1 can be detected co-localizing with DNA:RNA hybrids, which seem to contribute to its focal accumulation and/or its retention at DSBs. In particular, we also provide the first direct evidence that both the purified recombinant human BRCA1 and the constitutive BRCA1-BARD1 heterodimer can bind DNA:RNA hybrids in vitro, with an affinity similar to the dsDNA substrate. Additionally, the observation that dsDNA and

**Fig. 6** BRCA2 controls DNA:RNA hybrid levels at DSBs by interacting with RNase H2 and mediating its recruitment to DSBs. **a** Dot plot showing the normalized number of overlaps relative to random of γH2AX and RNASEH2A signals in S-phase cells knocked-down for BRCA2 and treated with NCS. At least $n = 50$ events were counted from two independent experiments. Lines represent mean ± s.e.m. **b** Representative images of PLA between RNASEH2A and γH2AX in not irradiated (no ir) or irradiated (2 Gy) HeLa-FUCCI cells knocked-down for BRCA2. Scale bar: 10 µm. Dot plot shows the number of signals per nucleus of PLA between RNASEH2A and γH2AX in cells knocked-down for BRCA2. At least $n = 150$ cells were counted from three independent experiments. Lines represent mean ± s.e.m. **c** Co-immunoprecipitation of endogenous RNASEH2A from not irradiated (no ir) or irradiated 5 Gy (ir) HEK293T cell extracts, prepared in the presence of benzonase to avoid contaminant nucleic acids. Asterisks indicate specific band. This experiment was repeated three times independently with similar results. **d** Immunoblot of GST pull-down assessing binding of recombinant Histidin (His)-tagged RNase H2 to GST-tagged BRCA2 fragments. This experiment was repeated two times independently with similar results. **e** Immunoblot of streptavidin pull-down assessing binding of recombinant Histidin (His)-tagged RNase H2 to biotinylated BRC repeats. This experiment was repeated two times independently with similar results. **f** Dot plot showing the normalized number of overlaps relative to random of γH2AX and DNA:RNA hybrid signals in S-phase cells knocked-down for BRCA2 and treated with NCS. At least $n = 50$ events were counted from two independent experiments. Lines represent mean ± s.e.m. **g** DRIP-qPCR at 1.5 kb on the right from the I-PpoI cut site within *DAB1* gene in S/G2-phase-sorted HeLa-FUCCI cells knocked-down for BRCA2 and transfected with the I-PpoI nuclease. The bar graph shows the average fold induction of cut samples relative to uncut from $n = 3$ independent experiments. Error bars represent s.e.m. $^*P < 0.05$, $^{***}P < 0.001$ (two-tailed Student's $t$ test). (RH2 is RNase H2; RH2A is RNASEH2A). Uncropped blots are shown in Supplementary Fig. 8. Source data are provided as a Source Data file

DNA:RNA hybrids compete for BRCA1-BARD1 binding suggests that both nucleic acids structures share the same binding site. This result provides evidence in support of a direct interaction between BRCA1 and DNA:RNA hybrids and it is consistent with the observed DNA:RNA hybrid-dependent BRCA1 recruitment to gene termination sites in living cells[42].

Recent evidence shows that excessive amounts of DNA:RNA hybrids at DSBs may dampen repair by HR, as demonstrated by impaired HR efficiency in the absence of the RNA-unwinding protein DDX1[29] or SENATAXIN[30]. In line with these results, in human and *Drosophila* cells, the catalytic component of the RNA exosome, which contributes to RNA degradation, localizes to DSBs and its activity is required for RAD51 loading[62]. In *S. pombe*, only controlled levels of DNA:RNA hybrids at DSBs facilitate repair[27]. Similarly, a precise level of DNA:RNA hybrids is required to guarantee the proper length of mammalian telomeres that elongate through an HR-based pathway named alternative lengthening of telomeres[63]. In mammalian cells, emerging links between HR proteins and DNA:RNA hybrids have recently been described and support our conclusions. DNA:RNA hybrids accumulate globally in cells lacking BRCA1 or BRCA2[43,44]. Additionally, proteins that, together with BRCA2, control the FA repair pathway localize to DNA damage sites via DNA:RNA hybrids[47,48]. At promoter regions, accumulation of DNA:RNA hybrids in BRCA2-deficient cells is due to reduced recruitment of RNA Pol II-associated factor 1 (PAF1), which promotes RNA Pol II release[46]. However, until now, no mechanisms explaining the emerging link between proteins controlling HR and DNA:RNA hybrid metabolism at DSBs have been proposed. Here, we provide the first evidence that RNase H2, the main protein responsible for DNA:RNA hybrid degradation in mammalian nuclei[57] localizes to DSBs during the S/G2 phase of the cell-cycle by directly interacting with BRCA2. Indeed, BRCA2 knock-down reduces RNase H2 recruitment while boosting DNA:RNA hybrid accumulation at DSBs. This observation is not only consistent, but it could help mechanistically explaining the increased DNA:RNA hybrid levels observed in BRCA2-depleted cells by others[43].

In summary, we propose a model (Fig. 7g) in which DNA:RNA hybrids form downstream of DNA-end resection, upon hybridization of dilncRNAs with resected DNA ends generated during the S/G2 cell-cycle phase. DNA:RNA hybrids are initially recognized by BRCA1 and, subsequently, BRCA2-mediated recruitment of RNase H2 induces their degradation, thus ensuring efficient HR-mediated repair.

## Methods

**Cell culture**. All the cell lines were grown under standard tissue culture conditions (37 °C, 5% $CO_2$). HeLa (ATCC, CCL-2) were grown in Minimum Essential Medium (MEM) (Biowest/Gibco) supplemented with 10% fetal bovine serum (FBS), 1% L-glutamine, nonessential amino acids (10 mM for each aa) and 1 mM sodium pyruvate; U2OS cells (ATCC, HTB-96) were grown in McCoy's 5A glutaMax (Gibco) supplemented with 10% FBS. HeLa-FUCCI (RIKEN BioResource Center cell bank)[51] and doxycycline-inducible I-SceI/DR-GFP (TRI-DR-U2OS) (kind gift from P. Oberdoerffer) were cultured in Dulbecco's Modified Eagle's medium (DMEM) (Lonza) supplemented with 10% FBS, 1% L-glutamine. For TRI-DR-U2OS cells, I-SceI expression was induced by adding 5 µg/ml doxycycline to the cell medium for 72 h. HEK293 TLR[sce] cells (kind gift from A.M. Scharenberg) were cultured in DMEM (Lonza) supplemented with 10% FBS, 1% L-glutamine, and 1 µg/ml puromycin. DIvA cells (AsiSI-ER-U20S) (kind gift from G. Legube) were cultured in DMEM without phenol red (Gibco) supplemented with 10% FBS, 1% L-glutamine, 1% pyruvate, 2.5% HEPES, and 1 µg/ml puromycin. AsiSI-dependent DSBs induction was obtained by treating the cells with 300 nM 4-hydroxytamoxifen (4-OHT) (Sigma-Aldrich) for 4 h.

U2OS cell synchronization for super-resolution imaging experiments was obtained by serum starvation. Briefly, cells were plated on glass coverslips for 24 h. G0/G1-phase synchronization was achieved by replacing complete medium with serum-free medium for 72 h. A mid-S phase cell population was obtained after 16 h release into complete medium. Double-strand breaks (DSBs) were generated by using the radiomimetic drug Neocarzinostatin (NCS) (Sigma-Aldrich).

All cell lines are tested for mycoplasma by PCR and by a biochemical test (MycoAlert, Lonza).

Ionizing radiation was induced by a high-voltage X-ray generator tube (Faxitron X-Ray Corporation).

**Transfections**. Two micrograms of GFP-RNase H1 plasmid (kind gift from N. Proudfoot) and its related control was transfected with Lipofectamine 2000 (Life Technologies) and experiments were performed 24 h after transfection.

One microgram of mammalian ER-I-PpoI (kind gift from M. Kastan) was transfected with Lipofectamine 2000. After 24 h transfection, nuclear translocation of ER-I-PpoI was induced by adding 4-OHT (Sigma-Aldrich) at 2 µM final concentration for the indicated time. When I-PpoI was transfected, medium without phenol red was used to avoid leakiness in the expression of the plasmid.

RNA interference was achieved by transfecting 5–20 nM siRNAs with Lipofectamine RNAiMAX transfection reagent (Life Technologies) for 48 h. For DNA:RNA hybrids detection upon EXO1, BRCA2, or RAD51 knock-down, cells were seeded and transfected with siRNAs in parallel. I-PpoI was transfected the day after. For DNA:RNA hybrids detection upon CtIP knock-down, cells were transfected with siCtIP for 48 h, then replated and transfected with I-PpoI. After 24 h, I-PpoI expression was induced by 4-OHT. Sequences of the siRNA used are listed in Supplementary Data 1.

For the DR-GFP experiments, TRI-DR-U2OS cells were transfected with a pool of antisense oligonucleotides (ASOs) (A1–A5 and B1–B5), which are mixmers containing locked nucleic acid oligonucleotides with a fully phosphorothioate backbone (Exiqon), at 20 nM final concentration using Lipofectamine RNAiMAX transfection reagent (Life Technologies). Concomitantly with ASOs transfection, I-SceI expression was induced by adding doxycycline to the cell media and HR was evaluated 72 h after, as described in the section "DR-GFP and TLR reporter assay".

For the TLR experiments, HEK293 TLR[sce] cells (kind gift from A. Scharenberg) were transfected overnight with a pool of ASOs (A2–A4 and B2–B4) at a 50 nM final concentration together with the IFP-tagged I-SceI (0.75 µg) and the BFP-tagged donor GFP template (0.75 µg) using Lipofectamine 2000 (Life Technologies). The IFP-tagged I-SceI, pRRL sEF1a HA.NLS.Sce(opt).T2A.IFP (Addgene plasmid # 31484), and the BFP-tagged GFP donor, pRRL SFFV d20GFP.T2A.mTagBFP Donor (Addgene plasmid # 31485), were a gift from A. Scharenberg[56]. The day after, medium containing the transfection reagents was

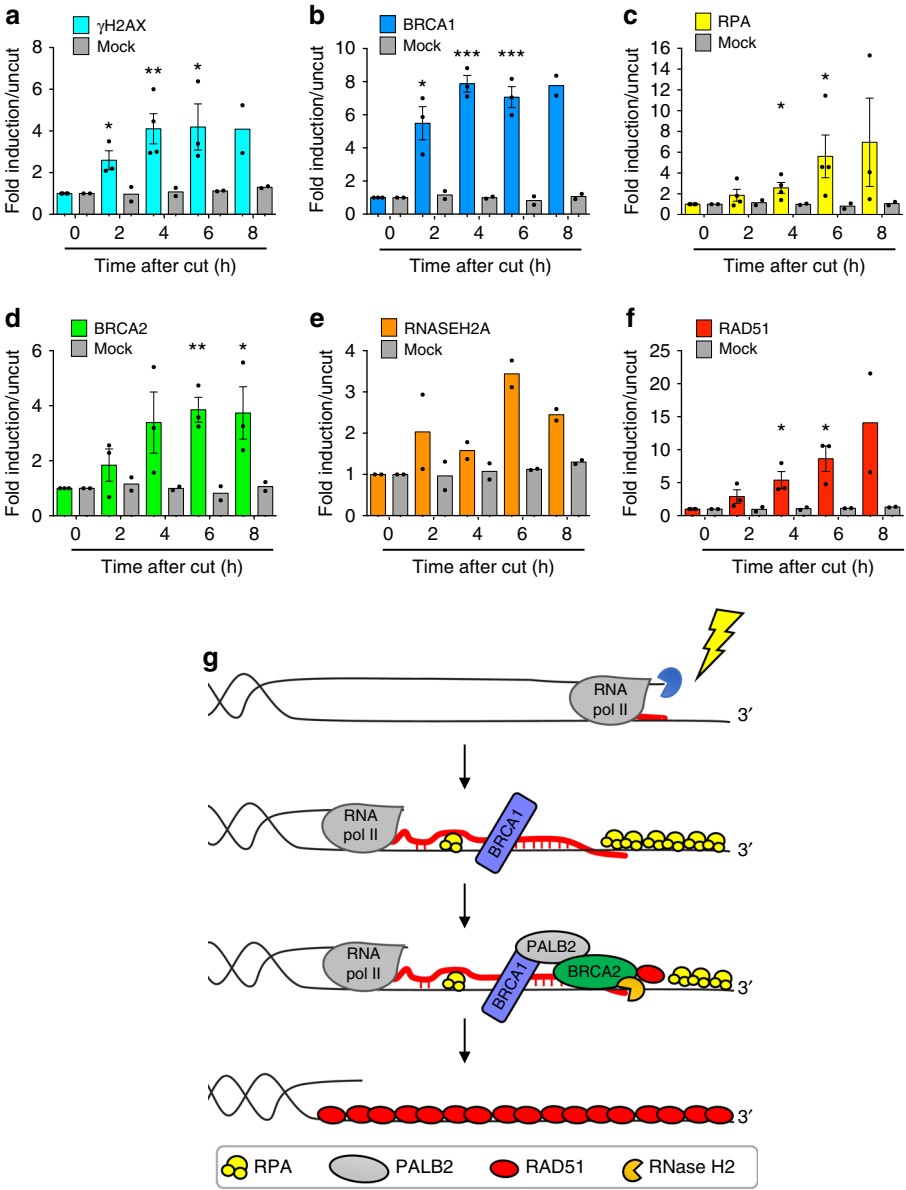

**Fig. 7** RNase H2 recruitment to DSBs occurs al later time-points after DSB induction. Bar graphs showing fold induction of **a** γH2AX, **b** BRCA1, **c** RPA, **d** BRCA2, **e** RNase H2, and **f** RAD51 signals at the nongenic AsiSI site analyzed in Fig. 1d. Lines represent mean ± s.e.m. ($n \geq 2$ biological replicates). $^{*}P < 0.05$, $^{**}P < 0.01$ (two-tailed Student's $t$ test). **g** Model of DNA:RNA hybrid dynamics at DSBs. Source data are provided as a Source Data file

replaced with fresh media and cells were harvested 3 days after. In both cases, before transfection, the ASO solution was incubated at 95 °C for 5 min and chilled on ice for 5 min to prevent the formation of secondary structures of the oligonucleotides. ASO sequences are listed in Supplementary Data 2.

**Inhibition of RNA polymerase II transcription**. RNA polymerase II transcription was inhibited by treatment with α-amanitin (50 μg/ml) (Sigma-Aldrich) or 5,6-dichloro-1-β-D-ribofuranosylbenzimidazole (DRB, 50 μM) (Sigma-Aldrich), respectively dissolved in deionized water and Dimethyl sulfoxide (DMSO). Immediately after adding the drugs, HeLa cells were irradiated (5 Gy) and then fixed 6 h later. For α-amanitin treatment, before adding the drug to the medium, cells were mildly permeabilized with 2% Tween 20 in phosphate-buffered saline (PBS) 1× for 10 min at room temperature (RT). RT-qPCR analysis of the levels of c-FOS, a short-lived RNA specifically transcribed by RNA pol II, was used to monitor the efficacy and specificity of the drugs.

**RNase H treatment**. U2OS cells were plated on coverslips and irradiated (2 Gy). One hour later cells were permeabilized with 0.2% Tween 20 in PBS 1× for 10 min at RT. After two washes in PBS 1×, each coverslip was incubated for 30 min at RT with 15 U RNase H (USB corporation) diluted in 200 μl PBS with 5 mM MgCl₂[64].

Next, coverslips were washed twice in PBS 1× and fixed and stained as described in the "Immunofluorescence and imaging analysis" section.

**RNA extraction**. Total RNA from cultured cells was extracted with Maxwell® RSC simplyRNA Tissue Kit with the Maxwell® RSC Instrument (Promega), according to manufacturer instructions.

For dilncRNAs detection, chromatin-bound RNA was extracted as follows. Cells were fractionated according to a published protocol[65]. The obtained chromatin fraction was treated with 50 U of Turbo DNase (Ambion) for 10 min at 37 °C and then digested with with 200 μg of Proteinase K (Roche) for 10 min at 37 °C. The RNA was then purified with Maxwell® RSC simplyRNA Tissue Kit.

**Standard RT-qPCR and strand-specific RT-qPCR**. For standard RT-qPCR, cDNA was obtained using the SuperScript VILO Reverse Transcriptase (Life Technologies), according to manufacturer instructions. Roche SYBR Green-based RT-qPCR experiments were performed on a Roche LightCycler 96 or 480 machine. RPPO was used as normalizer.

For dilncRNAs detection, chromatin-bound RNA was retro-transcribed using the Superscript IV First Strand cDNA synthesis kit (Invitrogen) with strand-specific primers. Expression of dilncRNAs was determined by RT-qPCR using EvaGreen Supermix (Bio-Rad). For dilncRNAs detection in DIvA cells, RT was

performed with the primer "AsiSI_non genic_1.5 Rev" and qPCR with primers "AsiSI_non genic_0.5". 7SK was used as normalizer. For dilncRNAs detection at the I-PpoI cut site, RT was performed with the primer "DAB1 + 1.5_Rev" and qPCR with primers "DAB1 + 1.5". 47S was used as normalizer. See Supplementary Data 3 for a complete list of primers used.

**Immunofluorescence and imaging analysis**. Cells were fixed in 4% paraformaldehyde (PFA) for 10 min at RT. For BRCA2 staining cells were fixed in ice-cold methanol for 10 min. Immunofluorescence was performed as described previously[17,18]. Secondary antibodies used were: goat anti-rabbit or anti-mouse Alexa 488 IgG (Life Technologies); donkey anti-mouse or anti-rabbit Cy3 IgG (Jackson Immuno Research), donkey anti-mouse or anti-rabbit Alexa 647 IgG (Life Technologies). For BrdU native staining cells were incubated with BrdU (Sigma-Aldrich, 10 μg/ml) for 24 h.

Immunofluorescence images were acquired using a widefield Olympus Biosystems Microscope BX71 and the MetaMorph software (Soft Imaging System GmbH). Confocal sections were obtained with a Leica TCS SP2 or AOBS confocal laser microscope by sequential scanning. Comparative immunofluorescence analyses were performed in parallel with identical acquisition parameters. Images were analyzed by CellProfiler 2.1.1 software[66].

**Super-resolution imaging**. Super-resolution (SR) experiments were performed on U2OS cells seeded on coverslips and synchronized as described in "Cell culture" section. Upon DNA damage induction, cells were pre-extracted at RT for 3 min in CSK buffer (10 mM Hepes, 300 mM sucrose, 100 mM NaCl, 3 mM MgCl₂, and 0.5% Triton X-100, pH 7.4) and fixed for 15 min in PFA (3.7% from 32% EM grade, Electron Microscopy Sciences, 15714) and glutaraldehyde (0.3% from 70% EM grade, Sigma-Aldrich, G7776) in PBS. Blocking was performed in blocking buffer (2% glycine, 2% bovine serum albumin (BSA), 0.2% gelatin, and 50 mM NH₄Cl in PBS) for 1 h at RT. Primary antibodies used are listed in Supplementary Data 4. Immediately before imaging analysis, coverslips were mounted onto a microscope microfluidics chamber and freshly prepared SR imaging buffer, comprising an oxygen scavenging system including 1 mg/ml glucose oxidase (SigmaAldrich, G2133), 0.02 mg/ml catalase (SigmaAldrich, C3155), 10% glucose (SigmaAldrich, G8270), and 100 mM mercaptoethylamine (Fisher Scientific, BP2664100) in PBS, was added to the imaging chamber. Images were acquired with a custom-built SR microscope based on a Leica DMI 3000 inverted microscope. For each field 2000 sequential frames of single molecule emissions at 40 Hz were collected and imaged on an electron-multiplying charged coupled device (EMCCD, Andor) using Solis software (Andor). Each raw image stack was processed for single molecule localization and rendered using 20 nm pixels via rapidSTORM or QuickPALM. Monte Carlo simulations were used to randomly rearrange the clusters within an ROI to calculate a baseline level of random colocalization. Using this approach, 20 random simulations were generated for each nucleus in a pair-wise fashion. The total number of overlaps detected in each nucleus (typically 15–100) was normalized to the determined random level of overlap by dividing the number of real overlaps by the average number of overlaps in the same randomly simulated nucleus. For display purposes, images were smoothed by applying a Gaussian blur filter and colors were thresholded for optimal production of a clear picture of the single foci.

**Proximity ligation assay**. Cells were labeled according to the manufacturer's instructions (Sigma). Briefly, cells were fixed as described in the section "immunofluorescence and imaging analysis" and incubated with primary antibodies overnight at 4 °C. Proximity ligation assay (PLA) probes (secondary antibodies conjugated with oligonucleotides) were added to the samples. After ligation of the oligonucleotide probes in close proximity (<40 nm), fluorescently labeled oligonucleotides were added together with a DNA polymerase to generate a signal detectable by a fluorescence microscope. Images were acquired using a widefield Olympus Biosystems Microscope BX71 and the MetaMorph software (Soft Imaging System GmbH) and quantification of PLA dots was performed with the automated image-analysis software CellProfiler 2.1.1.

**DR-GFP and TLR reporter assay**. TLR experiments were performed using HEK293 TLRˢᶜᵉ cells. IFP-tagged I-SceI and BFP-tagged donor GFP template were transfected together with ASOs, as described in "Transfections" section. After 72 h, the HR and mutagenic NHEJ efficiency was determined by flow cytometry analysis of the % of GFP- and mCherry- positive cells, respectively, on the gated IFP- and BFP-double-positive cell population, as described in the section "Fluorescence-activated cell sorting".

I-SceI expression in TRI-DR-U2OS cells was induced by adding 5 μg/ml doxycycline and 72 h later the HR efficiency was determined by quantifying GFP-positive cells (product of successful HR) by flow cytometry, as described in the section "Fluorescence-activated cell sorting". A PCR method was also used to monitor HR and SSA. DNA was extracted with the DNeasy Blood & Tissue Kit (Qiagen) and PCR was performed with the GoTaq® DNA Polymerase (Promega). HR was monitored by measuring the intensity of the amplicon generated by the primers P1 and P2[67,68]. SSA was monitored by measuring the intensity of the amplicon generated by amplification with primer F1 and R2[67,68]. Both the

measurements were normalized on the actin amplicon. Sequences of the primers are listed in Supplementary Data 3. Amplifications were performed using the following program and were determined to be in the linear range:

For P1-P2 primers: 98 °C/5 min × 1 cycle; 98 °C/45 s, 70 °C/30 s, and 72 °C/30 s × 30 cycles; 72 °C/10 min × 1 cycle.

For F1-R2 primers: 98 °C/45 s × 1 cycle; 98 °C/10 s, 50 °C/30 s, and 72 °C/60 s × 25 cycles; 72 °C/2 min × 1 cycle.

For Actin primers: 95 °C/5 min × 1 cycle; 95 °C/30 s, 62 °C/30 s, and 72 °C/30 s × 25 cycles; 72 °C/10 min × 1 cycle.

Totally, 10 μl of the PCR products was loaded on a 1% agarose gel and visualized by Gel Red staining. Images were acquired with Chemidoc imaging system (Bio-Rad) and densitometric analysis was performed using the Image Lab 5.2 software.

**Immunoblotting**. Cells were lysed in Laemmli sample buffer (2% sodium dodecyl sulfate (SDS), 10% glycerol, 60 mM Tris-HCl pH 6.8). Protein concentration was determined by the biochemical Lowry protein assay and the desired amount of protein was mixed with bromophenol blue and dithiothreitol (DTT), heated at 95 °C for 5 min, and resolved by SDS polyacrylamide gel electrophoresis (SDS-PAGE). After transferring on a nitrocellulose membrane (0.45 mm) (400 mA; 1 h) in transfer buffer (25 mM Tris-HCl, 0.2 M Glycine, 20% methanol), membranes were blocked with 5% milk in TBS-T buffer (Tween 20, 0.1%) for 1 h at RT and next incubated overnight at 4 °C with primary antibodies diluted in 5% milk in TBS-T (see Supplementary Data 4). Next, the membranes were washed with TBS-T three times for 10 min and incubated with secondary horseradish peroxidase (HRP)-conjugated antibodies (Bio-rad) diluted in 5% milk in TBS-T. After 3 more washes with TBS-T, HRP activity was detected using a Chemidoc imaging system (Bio-Rad) machine after adding the substrate for the enhanced chemiluminescent reaction ECL (GE Healthcare). Uncropped scans for main figure blots are shown in Supplementary Fig. 8.

**Immunoprecipitation**. HEK293T cells were collected and lysed in TEB150 lysis buffer (50 mM HEPES pH 7.4, 150 mM NaCl, 2 mM MgCl₂, 5 mM EGTA pH 8, 1 mM DTT, 0.5% Triton X-100, 10% glycerol, protease inhibitor cocktail set III (Calbiochem) and Benzonase 1:1000 (Sigma) for 45 min at 4 °C. Usually, 1 mg of the protein lysate was used per each immunoprecipitation in a reaction volume of 500 μl. As input, 1% of the immunoprecipitation reaction was collected and denatured in the sample buffer (50 mM Tris-HCl pH 6.8; 2% SDS; 10% glycerol; 12.5 mM EDTA; 0.02% bromophenol blue; 100 μM DTT) for 10 min at 95 °C. Unspecific binding of proteins to the beads was reduced by incubating samples with Protein G beads (50 μl) (Zymed Laboratories) for 1 h at 4 °C (preclearing). Binding reactions were performed overnight at 4 °C and were followed by addition of protein Glutathione–Sepharose beads for 2 h. After 3–6 washes with lysis buffer, immunoprecipitated proteins were released by the addition of sample buffer and incubation at 95 °C for 10 min.

**GST and streptavidin pull-down experiments**. GST and streptavidin pull-down experiments were performed as described in[58]. Recombinant human RNase H2 with a His-tag at the C-terminus of RNASEH2B (a gift from M. Reijns and A. Jackson) was purified as previously described[32]. Briefly, GST-BRCA2 fragments bound to Glutathione–Sepharose beads or biotin-BRC peptides bound to streptavidin beads were incubated with His-tagged RNase H2 in binding buffer (50 mM Tris-HCl (pH 7.4), 150 mM NaCl, 1% NP-40, 5 mM EDTA, 1 mM PMSF, 10 mM NaF, complete protease inhibitor cocktail (Roche)) for 20 min at RT. The beads were then extensively washed with the binding buffer and bound proteins were resolved with 4–12% MES before Western blotting.

**Chromatin immunoprecipitation**. ChIP was performed as described previously[17]. Briefly cells were cross-linked and chromatin was sonicated with a Focused-Ultrasonicator Covaris to obtained fragments of ~500 bp. 20 μg of chromatin were used per sample for γH2AX, BRCA1, and RAD51 ChIP; 50 μg for RPA ChIP, and 100 μg for RNASEH2A ChIP. DNA was cleaned up by QIAquick PCR purification column (Qiagen) according to the manufacturer's instructions and Roche SYBR Green-based qPCR experiments were performed on a Roche LightCycler 96 or 480 machine (see Supplementary Data 3 for primers sequence).

**DNA:RNA hybrids immunoprecipitation**. DRIP was performed following a published protocol[43]. Briefly, DNA was extracted gently with phenol:chloroform: isoamyl alcohol (Sigma-Aldrich cat. no. P2069) and digested with HindIII, EcoRI, BsrGI, XbaI, and SspI (NEB). After purification from restriction enzymes, half of the DNA was treated overnight with RNase H (NEB). In the meantime, serum-free medium containing the S9.6 antibody (kind gift from D. Piccini and M. Foiani) was mixed with protein A and protein G Dynabeads (Invitrogen) and incubated on a rotating wheel overnight at 4 °C. After a further purification step, 4 μg of DNA was used for each IP. After elution from the beads, DNA was cleaned up with QIAquick PCR purification column (Qiagen) according to the manufacturer's instructions. The indicated regions were amplified by Roche SYBR Green-based qPCR on a Roche LightCycler 480 machine (see Supplementary Data 3 for primers sequence).

The signal intensity plotted is the relative abundance of DNA–RNA hybrid immunoprecipitated in each region, normalized to input values.

**Fluorescence-activated cell sorting.** For DR-GFP and TLR experiments, cells were fixed in 1% formaldehyde for 20 min on ice. Next, cells were washed in PBS with 1% BSA and fixed in 75% ethanol. Fixed cells were washed again in PBS with 1% BSA and stained with propidium iodide (PI) (Sigma-Aldrich, 50 μg/ml) in PBS supplemented with RNase A (Sigma-Aldrich, 250 μg/ml). For cell cycle analysis, cells were directly fixed in 75% ethanol, as described above. Samples were acquired on an Attune NxT machine and analyzed with FlowJo_V10 software. At least $10^4$ events were analyzed per sample.

For sorting of HeLa-FUCCI cells, cells were collected in PBS with 2% FBS and the G1 and S/G2 population were sorted with a MofloAstrios (Beckman Coulter) in PBS supplemented with RNaseOUT (Thermo Fisher). Sorted samples were processed for DRIP as described in the section "DNA:RNA hybrids immunoprecipitation".

**Cloning, expression, and purification of recombinant proteins.** Recombinant BRCA1 was expressed and purified as a complex in *Sf*9 cells by co-infection with baculoviruses prepared from individual pFastBac1 plasmids pFB-2xMBP-BRCA1-10×His. Bacmids, primary and secondary baculoviruses were obtained using standard procedures according to manufacturer's instructions (Bac-to-Bac, Life Technologies). *Sf*9 cells were transfected using a Trans-IT insect reagent (Mirus Bio).

For the large-scale BRCA1 expression and purification, *Sf*9 cells were seeded at $0.5 \times 10^6$ per ml and infected 16 h later with recombinant baculoviruses expressing pFB-2×MBP-BRCA1-10×His. The infected cells were incubated in suspension at 27 °C for 52 h with constant agitation. All purification steps were carried out at 4 °C or on ice. The *Sf*9 cell pellets were resuspended in three volumes of lysis buffer (Tris-HCl, pH 7.5, 50 mM; Dithiothreitol (DTT), 1 mM; ethylenediaminetetraacetic acid (EDTA), 1 mM; protease inhibitory cocktail, Sigma P8340, 1:400; phenylmethylsulfonyl fluoride (PMSF), 1 mM; leupeptin, 30 μg/ml; NP40, 0.5%) for 20 min with continuous stirring. Glycerol was added to 16% (v/v) concentration. Next, 5 M NaCl was added slowly to reach a final concentration of 305 mM. The cell suspension was further incubated for 30 min with continuous stirring, centrifuged at 57,800 g for 30 min to obtain soluble extract. Pre-equilibrated amylose resin (New England Biolabs) was added to the cleared soluble extract and incubated for 1 h with continuous mixing. The resin was then collected by centrifugation at 2000 g for 2 min and washed extensively batch wise as well as on disposable columns (Thermo Scientific) with wash buffer (Tris-HCl, pH 7.5, 50 mM; β-mercaptoethanol, 2 mM; NaCl, 300 mM; glycerol, 10%; PMSF, 1 mM; NP40, 0.5%). Protein was eluted with wash buffer containing 10 mM maltose (Sigma). The eluates were further treated with PreScission protease for 90 min to cleave off the maltose binding protein affinity tag (MBP). The sample was then supplemented with 20 mM imidazole and further incubated with pre-equilibrated Ni-NTA agarose resin (Qiagen) for 1 h. The Ni-NTA resin was transferred on a disposable column and washed extensively with Ni-NTA wash buffer (Tris-HCl, pH 7.5, 50 mM; β-mercaptoethanol, 2 mM; NaCl, 1 M; glycerol, 10%; PMSF, 1 mM; imidazole, 20 mM). Prior to elution, the protein was washed once with the same Ni-NTA wash buffer listed above but with only 150 mM NaCl. Pooled fractions were stored at −80 °C.

**Electrophoretic mobility shift assay.** The 50 bp-long dsDNA substrate was prepared by annealing oligonucleotides X12-3 (5′GACGTCATAGACGATTA CATTGCTAGGACATGCTGTCTAGAGACTATCGC3′) and X12-4C (5′GCGA TAGTCTCTAGACAGCATGTCCTAGCAATGTAATCGTCTATGACGTC 3′′) as previously described[69]. The 50 bp-long DNA:RNA hybrid substrate was prepared by annealing X12-3 DNA and X12-4C RNA. BRCA1 was incubated for 30 min at 37 °C with 10 nM dsDNA or DNA:RNA hybrid substrate in a reaction buffer containing Tris-HCl pH 7.5, 25 mM; KCl, 90 mM; EDTA pH 8.0, 1 mM; DTT, 1 mM; BSA (NEB), 0.1 mg/ml; RNaseOUT (Invitrogen).

The cold competitor experiments were done by pre-binding BRCA1-BARD1 to 1 nM radiolabeled dsDNA or DNA:RNA at 37 °C for 10 min. Cold dsDNA or DNA:RNA was then added to the reaction and incubated for another 10 min before analyzing products on a gel. The salt titration experiments were done using the same conditions with the given salt concentrations. After the binding reaction, loading dye (5 μL; 50% glycerol; Tris-HCl pH 7.5, 20 mM; EDTA, 0.5 mM; bromophenol blue) was added to reactions and products were separated by 6% non-denaturing polyacrylamide gel electrophoresis at 4 °C. The gels were dried on 17CHR filter paper (Whatman), exposed to storage phosphor screens, and scanned by a Typhoon Phosphor imager (FLA 9500, GE Healthcare). Quantification of protein-bound substrates was done using Imagequant software.

**DRIP-seq and BLISS data analysis.** DRIP-seq raw FASTQ data were retrieved from the European Nucleotide Archive public repository (https://www.ebi.ac.uk/ena/data/view/PRJEB24001). Alignment was performed as described in ref. [30]: raw reads were aligned to the human genome (GRCh37/hg19 assembly) using BWA[70], and aligned reads were processed and duplicates were removed using SAMtools[71]. The log2 ratio of the fold enrichment of cut *vs* uncut DRIP-seq samples, was calculated using bamCompare (settings: --binSize 50 --normalizeUsing RPKM

--outFileFormat bedgraph) from the suite DeepTools[72]. To calculate the distribution profile of DNA:RNA hybrids enrichment at DSBs, the log2 ratio values calculated for each 50 bp bin for DRIP-seq reads were intersected to five distance intervals from the top 50 cut AsiSI sites (determined in our laboratory[50]): 0–0.5, 0.5–1, 1–1.5, 1.5–2, and 2–2.5 kb requiring a minimum overlap of 90% of the bin using intersectBed tool from BedTools suite[73]. For each AsiSI site a mean log2 ratio value was calculated with ad hoc bash scripts among the ones of the bins overlapping the site. The accumulation of DNA:RNA hybrids upon DSB induction was then measured separately at genic or non-genic AsiSI sites Data parsing, plotting, and statistical test calculation was performed using ad hoc R scripts and publicly available R packages.

**Statistical analysis.** Prism 6 software was used to generate graphs, to perform statistical analysis and to remove outliers with the Robust regression and Outlier removal method in Figs. 5e, 6b, and Supplementary Fig. 6d–f. Statistical analysis was performed with the two-tailed Student's *t* test, unless differently indicated. Asterisks in the figures indicate *P* value: *$P < 0.05$, **$P < 0.01$, ***$P < 0.001$, ****$P < 0.0001$.

**Reporting summary.** Further information on experimental design is available in the Nature Research Reporting Summary linked to this article.

## Data availability
All relevant data are available from the authors upon reasonable request. A reporting summary for this Article is available as a Supplementary Information file. Source data are provided as a Source Data file. DRIP-seq raw FASTQ data were retrieved from the European Nucleotide Archive public repository (https://www.ebi.ac.uk/ena/data/view/PRJEB24001).

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

## Acknowledgments

We thank M. Reijns and A. Jackson (MRC Human Genetics Unit, Institute of Genetics and Molecular Medicine, University of Edinburgh, Edinburgh, UK) for providing recombinant RNase H2 and for advice. We thank G. Legube (Centre de Biologie Inté-grative, Toulouse, France), M. Kastan (Duke Cancer Institute, Durham, USA), N. Proudfoot (Sir william dunn school of pathology, University of Oxford), and B. Xia (The Cancer Institute of New Jersey, University of Medicine and Dentistry of New Jersey, USA) for reagents and all F.d'A.d.F. group members for reading the manuscript, support and constant discussions. We also thank IFOM Cell Biology Unit, Kitchen Unit and Imaging TDU. G.D. was supported by Fondazione Italiana Ricerca Sul Cancro (Appli-cation No. 15050). C.J.-W. is supported by Fondazione Italiana Ricerca Sul Cancro (Application No. 19589). Work in the Rothenberg's laboratory is supported by grants from the NIH (CA187612, GM108119) and the American Cancer Society (RSG DMC-16-241-01-DMC). Work in Petr Cejka's laboratory is supported by European Research Council (681630) and the Swiss National Science Foundation (31003A_175444). Work in Ashok Venkitaraman's laboratory was supported by Medical Research Council (MRC) core awards MC_UU_12022/1 and MC_UU_12022/8. Work in Fabrizio d'Adda di

Fagagna's laboratory is supported by the Associazione Italiana per la Ricerca sul Cancro, AIRC (application 12971), Cariplo Foundation (grant 2010.0818 and 2014-0812), Fondazione Telethon (GGP12059 and GGP17111), Association for International Cancer Research (AICR-Worldwide Cancer Research Rif. N. 14-1331), Progetti di Ricerca di Interesse Nazionale (PRIN) 2010–2011 and 2015, the Italian Ministry of Education Universities and Research EPIGEN Project, a European Research Council advanced grant (322726), AriSLA (project 'DDRNA and ALS'), AIRC Special Program 5 per mille metastases Project n 21091", AMANDA project Accordo Quadro Regione Lombardia–CNR and flagship progetto InterOmics.

## Author contributions

D.W., W.L, and M.M performed super-resolution imaging and D.W. performed analyses; E.R. supervised their work. S.H. performed EMSA with purified recombinant proteins and P.C. supervised the work. X.R. performed the GST-BRCA2 and streptavidin pull-downs with M.L.'s assistance and A.V. supervised their work and analyzed the results. V.V. performed experiments to detect dilncRNAs. M.A. performed immunoprecipitation experiments; C.J.-W. performed the experiments with the TLR reporter system. G.D. and C.J.-W. performed the experiments with the DR-GFP system. F.I. performed bioinformatics analyses. V.M. contributed with technical support. G.D. generated all remaining data and wrote the manuscript; F.d'A.d.F. supervised the project and revised the manuscript; all authors edited the manuscript.

## Additional information

**Competing interests:** The authors declare no competing interests.

