## [Peer Review File · Nature Communications]

Editorial Note: This manuscript has been previously reviewed at another journal that is not operating a transparent peer review scheme. This document only contains reviewer comments and rebuttal letters for versions considered at Nature Communications .

REVIEWERS' COMMENTS:

Reviewer #1 (Remarks to the Author):

The revised manuscript is much improved. They have now provided additional experimental data to characterize DNA:RNA hybrids at DSBs and provided new evidence to show that BRCA1 preferentially binds to DNA:RNA hybrids. They have also conducted new experiments to strengthen their conclusion that BRCA2 mediates RNase H2 recruitment to DSBs. Their RNase H2A knockdown experiment could not lead to clear conclusion because they found that knockdown of RNase H2A affected expression of HDR proteins. However, this minor setback did jeopardize their overall conclusion. Therefore, the current form of the manuscript may be a suitable contribution to Nature Communications.

Minor point:

The authors have done statistical analysis on most quantification data. It is necessary to indicate the p value on Figure 4b, supplemental figure 2a, 2b-e, 3a, 3b, 5a-c, 6g-l, and 7a-f, even though they are obviously significant or not significant.

REVIEWERS' COMMENTS:

Reviewer #1 (Remarks to the Author):

The revised manuscript is much improved. They have now provided additional experimental data to characterize DNA:RNA hybrids at DSBs and provided new evidence to show that BRCA1 preferentially binds to DNA:RNA hybrids. They have also conducted new experiments to strengthen their conclusion that BRCA2 mediates RNase H2 recruitment to DSBs. Their RNase H2A knockdown experiment could not lead to clear conclusion because they found that knockdown of RNase H2A affected expression of HDR proteins. However, this minor setback did jeopardize their overall conclusion. Therefore, the current form of the manuscript may be a suitable contribution to Nature Communications.

Minor point:

The authors have done statistical analysis on most quantification data. It is necessary to indicate the p value on Figure 4b, supplemental figure 2a, 2b-e, 3a, 3b, 5a-c, 6g-l, and 7a-f, even though they are obviously significant or not significant.

We thank the referee for his/her comments.

P-value is shown in Figure 4b. Supplementary Figures 2a,c and 3a,b are qPCRs experiments shown as one representative experiment, as indicated in the figure legends. Since no statistics should be added in this case, we are prone to leave the text unaltered.

In regarding to the request of adding p-values even if when there is not statistical significance, this seems not be consistent with other papers published in Nat Comms and, therefore, also in this case we would leave things unaltered.